# How Does Local Landscape Geometry Evolve in Language Model Pre-Training?

## Abstract

The scale and expense of pre-training language models make efficient hyperparameter tuning essential, yet a principled guidance is still missing. Recent work shows that the geometry of loss landscape shapes training dynamics of neural networks and further informs hyperparameter choices. In this work, we analyze language model pre-training dynamics from a local landscape geometry perspective. Our study reveals two distinct phases. In the *early* phase, sharpness of the local landscape is initially high, leading to instability and loss plateaus under large learning rates (LRs). Later, the landscape shifts from sharp to flatter regions. This dynamic explains the necessity of LR warmup and further suggests that larger peak LRs require proportionally longer warmup periods. In the *late* phase, the local landscape is governed by the gradient noise scale. Through diffusion-limit analysis, we prove a *depth–flatness trade-off*: high noise from smaller batches widens the loss basin, whereas reduced noise from larger batches deepens it. This theory motivates a dynamic batch-size (BS) scheduler that begins with a small BS and increases it late in training. Together, we provide a unified account of loss landscape evolution, which translates into actionable tuning strategies for large-scale pre-training.

## 1 Introduction

Training language models efficiently requires carefully tuned hyperparameters, yet a principled guidance for tuning remains unclear. While practitioners often rely on grid search or trial-and-error, these approaches are costly and unreliable at scale. Recent research (Foret et al., 2021; Cohen et al., 2021; Gilmer et al., 2022) has highlighted that the geometry of the local loss landscape offers fundamental insights into optimization, revealing how factors such as sharpness[1] (Keskar et al., 2017; Zhang et al., 2017; Jiang et al., 2020) interact with hyperparameters to shape training dynamics. Consequently, leveraging insights from the local landscape geometry presents a promising path toward principled hyperparameter tuning for language model pre-training.

Several pioneering works have already attempted to study language models from the local landscape geometry perspective. Zhang et al. (2024a); Wang et al. (2025) identified blockwise sharpness patterns in language models through Hessian-based analyses. Wen et al. (2024) introduced the "river-valley" landscape to explain the effectiveness of Warmup-Stable-Decay (WSD) schedules (Hu et al., 2024). Peng et al. (2024); Chen et al. (2025) further visualized the loss landscapes of finetuned language models, offering geometric insights into the safety alignment. However, few studies have investigated the *dynamics* of local landscape geometry during language model pre-training.

To this end, we pose the central research questions of this paper:

1. *How does the local landscape geometry evolve in language model pre-training?*
2. *What implications does this evolution have for principled hyperparameter tuning?*

**Our contributions.** In this work, we present the first systematic study of the evolution of local landscape geometry during language model pre-training. As illustrated in Figure 1, our analysis reveals two distinct phases, each with significant implications for hyperparameter tuning.

• *Early in Training: From Sharp to Flat Landscapes.* In the early phase, we observe that the model shifts from sharper regions of the loss landscape toward flatter ones, contrary to the progressive

---

[1]To avoid misunderstanding, we clarify the terminology in Table 1.

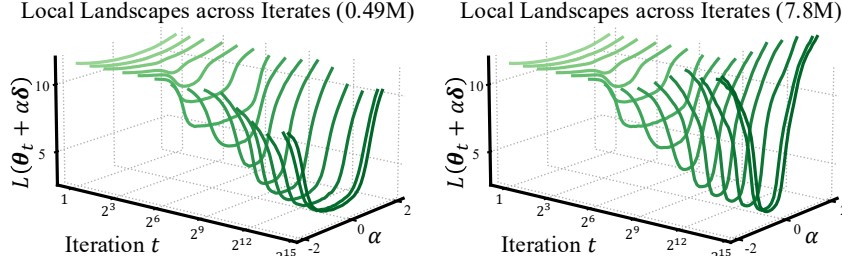

Figure 1: **The evolution of local loss landscape throughout pre-training.** We train `LLaMA-2` models with 170M parameters using different BSs (0.49M and 7.8M), and visualize the one-dimensional loss landscape at iterate $\boldsymbol{\theta}_t$ along a random direction $\boldsymbol{\delta}$, i.e., plot $L(\boldsymbol{\theta}_t + \alpha\boldsymbol{\delta})$ vs. the perturbation coefficient $\alpha$. The landscapes are shown across different training iterations $t$. **Early phase.** The landscapes gradually widen/flatten for both training runs. **Late phase.** Training with smaller BS produces wider landscapes than training with larger BS.

sharpening phenomenon in prior works (Cohen et al., 2021; Song & Yun, 2023; Cohen et al., 2025). Lyapunov stability analysis in Section 4 shows that the maximum stable learning rate (LR) is inversely proportional to sharpness. Since sharpness is extremely high early in pre-training, using large peak LRs without sufficient warmup leads to instabilities, such as loss spikes and plateaus (see Figure 2).

**Implications.** The sharp-to-flat transition explains the necessity of LR warmup: LR should remain small until sharpness has sufficiently decayed, preventing training instabilities. This further provides a practical tuning recipe: within a reasonable range, larger peak LRs require proportionally longer warmup, to safely navigate the sharpest stage of training.

• *Late in Training: Basin Selection Governed by Noise Scale.* In the late phase, the local landscape geometry is largely governed by the noise scale during training, with batch size (BS) $B$ serving as its primary controller. Our analysis shows that smaller BS widens the loss basin, while larger BS deepens it. Theoretically, we analyze the diffusion limit of preconditioned SGD, which uncovers a depth–flatness trade-off: reduced gradient noise tends to minimize the loss, leading to deeper minima; whereas increased noise tends to regularize the sharpness of landscape, moving toward wider ones.

**Implications.** The trade-off, together with the extensive ramping-time experiment in Figure 6, motivates a principled BS scheduling strategy: begins with a small BS and ramps it until the late phase of training. Our scheduling ensures steady loss reduction with minimal token consumption, ultimately achieving lower terminal loss than constant-BS training. Moreover, since the noise scale is proportional to $\eta/B$ in our theory, we predict that BS ramping and LR decay reduce the noise scale in similar ways and thus yield comparable performance (see Figure 8).

In summary, our work provides a two-phase picture of landscape evolution in pre-training: an early sharp-to-flat transition that necessitates LR warmup, and a late noise-driven regime that motivates BS scheduling. This unified view advances our understanding of pre-training dynamics and underscores the importance of landscape geometry in offering principled guidance for hyperparameter tuning.

## 2 RELATED WORKS

**Local Landscape Geometry (Sharpness) Evolution.** Understanding how local landscape geometry, particularly sharpness, evolves during training has drawn significant attention before the success of large language models. Wu et al. (2018); Cohen et al. (2022); Song & Yun (2023); Cohen et al. (2025) showed that initially gradient descent (GD) tends to move from flatter to sharper regions of the landscape. In addition, Jastrzębski et al. (2019); Jastrzebski et al. (2020) argued that in SGD, sharpness also changes monotonically but either increase or decrease depending on the setting. In the later phase, however, sharpness is largely governed by the properties of the optimizer (Zhou et al., 2025). One notable example is that the stochastic noise introduced by SGD and its variants implicitly biases training toward flat minima (Wu et al., 2018; Zhu et al., 2019; Xie et al., 2021; Wu et al., 2022). Yet, these findings are largely restricted to small-scale networks; *In comparison*, our work presents the *first systematic study* of how local landscape geometry evolves in large-scale language model pre-training, offering new insights into LR warmup and the design of BS schedules.

**Large-Scale Pre-training: Learning Rate Warmup.** Learning rate warmup, first introduced in large-batch ResNet (He et al., 2016; Goyal et al., 2017) and Transformer training (Vaswani et al., 2017), is now standard in large-scale pre-training (Shoeybi et al., 2019; Zhang et al., 2022; Hu et al., 2024). Its mechanism, however, remains only partly understood. Gotmare et al. (2019) showed that warmup prevents excessively large early parameter updates; Bergsma et al. (2025) attributed the early updates to bias reduction, rather than curvature. Gilmer et al. (2022) argued that warmup guides optimization into flatter regions where large LRs are stable; and Kosson et al. (2024) showed in language model pre-training that warmup mitigates momentum bias correction and correlated gradients that otherwise drive unstable representation shifts. Yet no unified explanation exists. *In comparison*, our work views warmup from a *unified* geometric perspective, suggesting that larger peak LRs demand proportionally longer warmup.

**Large-Scale Pre-training: Batch Size Schedules.** Batch size is another critical hyperparameter in large-scale pre-training, shaping the trade-off between step efficiency and data efficiency. Most prior work (McCandlish et al., 2018; Kaplan et al., 2020; Gray et al., 2023; 2024; Zhang et al., 2025) has focused on the critical batch size (CBS), the point where further increasing BS yields diminishing returns. However, CBS is typically treated as a constant, and much less attention has been given to *BS scheduling*. Early works on adaptive sampling proposed gradually increasing BS to balance efficiency and noise reduction (De et al., 2017; Lau et al., 2024b;a; 2025; Ostroukhov et al., 2024). However, these studies remain mostly theoretical. Advanced language models (Brown et al., 2020; Touvron et al., 2023; Liu et al., 2024; Li et al., 2025) employed stage-wise BS schedules, but without systematic analysis. *In contrast*, our work connects BS scheduling to the evolving local landscape geometry, providing a principled foundation for when and how to expand BS during pre-training.

## 3 PRELIMINARIES

**Basic Notations.** We use bold lowercase letters (e.g., $\boldsymbol{x} = (x_i)$) to denote vectors and bold uppercase letters (e.g., $\mathbf{A} = (a_{ij})$) to denote matrices. For a matrix $\mathbf{A}$, let $\|\mathbf{A}\|_2$, $\|\mathbf{A}\|_F$, and $\mathrm{Tr}(\mathbf{A})$ denote its spectral norm, Frobenius norm and trace, respectively. The Hadamard product is denoted by $\odot$.

**Theoretical Setup.** Our theory focuses on the preconditioned stochastic gradient descent (PSGD). We consider a model with parameters $\boldsymbol{\theta} \in \mathbb{R}^p$ and a training set of $n$ examples. Let $L_i(\boldsymbol{\theta})$ be the fitting error evaluated at the $i$-th example and $L(\boldsymbol{\theta}) = \frac{1}{n}\sum_{i=1}^{n} L_i(\boldsymbol{\theta})$ be the empirical risk. We analyze the preconditioned SGD with a fixed positive-definite[2] preconditioner $\mathbf{M} \succ 0$. At iteration $k$, the update rule gives:

$$\boldsymbol{\theta}_{k+1} = \boldsymbol{\theta}_k - \eta \mathbf{M}(\nabla L(\boldsymbol{\theta}_k) + \boldsymbol{\xi}_k), \tag{1}$$

where $\eta > 0$ is the LR and $\{\boldsymbol{\xi}_k\}$ are i.i.d. random noise vectors with

$$\mathbb{E}[\boldsymbol{\xi}_k] = \mathbf{0}, \quad \mathbb{E}[\boldsymbol{\xi}_k \boldsymbol{\xi}_k^\top] = \boldsymbol{\Sigma}(\boldsymbol{\theta}_k)/B. \tag{2}$$

Note that $\boldsymbol{\Sigma}(\boldsymbol{\theta}_k) = \frac{1}{n}\sum_{i=1}^{n} \nabla L_i(\boldsymbol{\theta}_k)\nabla L_i(\boldsymbol{\theta}_k)^\top - \nabla L(\boldsymbol{\theta}_k)\nabla L(\boldsymbol{\theta}_k)^\top$ is the gradient covariance at $\boldsymbol{\theta}_k$, and $B$ denotes the BS. During the late phase of training, the model remains close to some global minimum $\boldsymbol{\theta}^\star$ and the loss can be approximated quadratically:

$$L(\boldsymbol{\theta}) = L(\boldsymbol{\theta}^\star) + \frac{1}{2}(\boldsymbol{\theta} - \boldsymbol{\theta}^\star)^\top \mathbf{H}(\boldsymbol{\theta}^\star)(\boldsymbol{\theta} - \boldsymbol{\theta}^\star), \quad \mathbf{H}(\boldsymbol{\theta}^\star) := \nabla^2 L(\boldsymbol{\theta}^\star) \succ 0. \tag{3}$$

Similar formulations have been widely used in dynamical stability analyses (Wu et al., 2018; Cohen et al., 2021; Zhou et al., 2025) and theoretical advances on BS scaling (McCandlish et al., 2018).

**Experimental Setup.** Our experiments are mainly conducted on LLaMA-2 architecture (Touvron et al., 2023) models with 93M and 170M parameters. Training is performed on the FineWeb-Edu dataset (Penedo et al., 2024), with sufficient training budgets ranging from 50 to 1000 tokens-per-parameter (TPP)[3] and a context length of 1024. We adopt AdamW (Kingma & Ba, 2014) with hyperparameters $\beta_1 = 0.95$, $\beta_2 = 0.95$, and weight decay 0.1, together with gradient clipping at 1.0 for stability. The evaluation is conducted on a held-out validation split of approximately 50M tokens. More experiments on larger scales, other architectures, and optimizers are deferred to Section D.

---

[2]Most practical preconditioners are positive-definite: $\mathbf{M} = I$ for SGD, diagonal $\mathbf{M}$ for AdaGrad (Duchi et al., 2011), RMSProp (Tieleman & Hinton, 2012), Adam, etc.

[3]At least $10\times$ over Chinchilla-optimal tokens (Hoffmann et al., 2022).

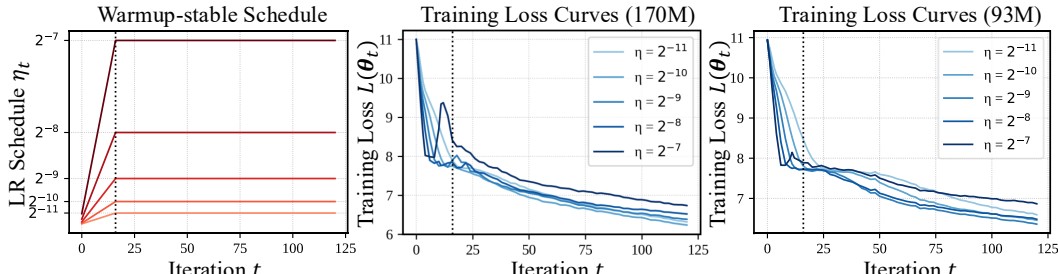

Figure 2: **Loss spikes and plateaus early in training.** We train a series of LLaMA-2 models with 93M and 170M parameters. We adopt a warmup-stable schedule, where the warmup length is shortened to 16 iterations and the peak LR is varied, $\eta \in \{2^{-11}, 2^{-10}, 2^{-9}, 2^{-8}, 2^{-7}\}$. **(Left).** LR schedule: $\eta_t$ vs. training iteration $t$. **(Middle, Right).** Training loss curves for different model sizes: $L(\boldsymbol{\theta}_t)$ vs. training iteration $t$. The vertical dashed line marks the end of the warmup phase.

Our experiments vary the LRs and BSs. In Section 4, we primarily study the role of LR and warmup length, fixing BS at 7.8M. In Section 5, we focus on the effect of BS, with LR fixed at $2^{-10}$. To decouple BS ramping from LR decay, we adopt a *warmup-stable* schedule: after linear warmup to the peak value, the LR remains constant (similar to WSD (Hu et al., 2024)), but without decay phase.

## 4 EARLY IN PRE-TRAINING: FROM SHARP TO FLAT LANDSCAPES

In this section, we provide evidence that, during the early phase of pre-training, the local landscape of language models evolves from sharp regions toward flatter ones. We first observe that training with large LRs and insufficient warmup often leads to instability and early loss plateaus. By Lyapunov stability analysis, we then attribute these behaviors to sharp-to-flat dynamics occurring in the initial phase of training. This finding explains why pre-training needs LR warmup and suggests that larger peak LRs require proportionally longer warmup periods.

**Motivating Observations: Instability and Loss Plateaus Early in Training.** The loss curves for pre-training are typically smooth initially; the model escapes from random initialization and the loss decreases rapidly. Yet, surprisingly, when the warmup length is extremely shortened, we *consistently* observe loss spikes and plateaus near the end of the warmup phase.

To demonstrate this, we train models of different sizes with a fixed warmup length of 16 iterations while varying the peak LR. As shown in Figure 2, a loss plateau reliably appears around the end of the warmup phase across all settings. Additionally, larger LRs produce higher spikes, which mark a characteristic feature of early training instability. Given these results, two natural questions arise:

**Q1.** *Why does shortened warmup induce training instability?*

**Q2.** *Why do spikes and plateaus occur only at the very beginning of training?*

To shed light on these questions, we analyze the dynamics of PSGD via Lyapunov stability analysis.

**Lyapunov Stability Analysis: Sharpness Matters.** Let $\boldsymbol{\theta}_k, \tilde{\boldsymbol{\theta}}_k$ be two nearby trajectories, and define their difference as $\boldsymbol{e}_k := \tilde{\boldsymbol{\theta}}_k - \boldsymbol{\theta}_k$. When the noise term $\boldsymbol{\xi}$ is set to zero, the evolution of $\boldsymbol{e}_k$ satisfies:

$$\boldsymbol{e}_{k+1} = \boldsymbol{e}_k - \eta \mathbf{M}(\nabla L(\boldsymbol{\theta}_k + \boldsymbol{e}_k) - \nabla L(\boldsymbol{\theta}_k)) \overset{\text{(Linearization)}}{=} (\mathbf{I} - \eta \mathbf{M}\mathbf{H}(\boldsymbol{\theta}_k))\boldsymbol{e}_k, \quad (4)$$

The dynamics in Equation (4) describe the local sensitivity of the iteration: if matrix $(\mathbf{I} - \eta \mathbf{M}\mathbf{H}(\boldsymbol{\theta}_k))$ repeatedly expands $\boldsymbol{e}_k$, small perturbations grow exponentially and the iterates are linearly unstable. Intuitively, the LR $\eta$ interacts directly with the curvature of the landscape: if $\eta$ is too large relative to the sharpest direction, the update rule amplifies perturbations and leads to loss spikes. The following lemma formalizes this stability condition for preconditioned GD.

**Lemma 4.1** (Stability Condition for Preconditioned GD). *Define the preconditioned curvature matrix* $\mathbf{S}(\boldsymbol{\theta}_k) := \mathbf{M}^{1/2}\mathbf{H}(\boldsymbol{\theta}_k)\mathbf{M}^{1/2}$, *and let* $\{\lambda\}_{i=1}^p$ *be the eigenvalues of* $\mathbf{S}(\boldsymbol{\theta}_k)$. *The linear system in Equation* (4) *is asymptotically stable (i.e.,* $\lim_{k\to\infty} \boldsymbol{e}_k = \mathbf{0}$*) if* $\eta$ *satisfies* $0 < \eta < \frac{2}{\lambda_{\max}(\mathbf{S}(\boldsymbol{\theta}_k))}, \forall k \geq 0$.

Lemma 4.1 shows that the Lyapunov stability is governed by the largest eigenvalue of $\mathbf{S}$. If the curvature along the sharpest direction is too large, only a sufficiently small LR can prevent divergence.

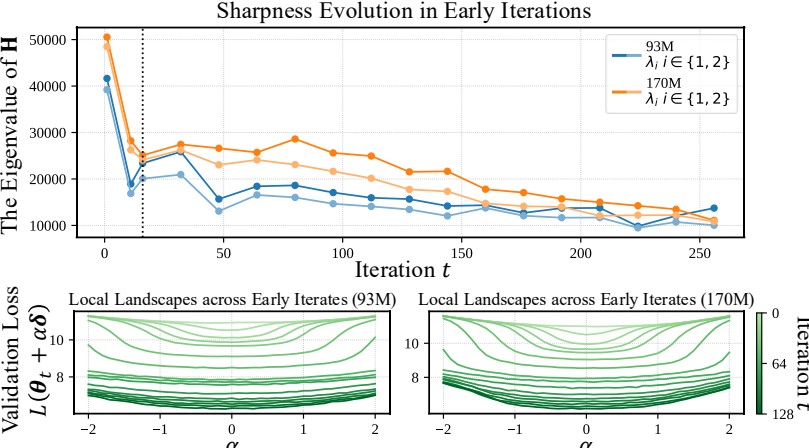

Figure 3: **Early pre-training shifts iterates from sharp to flat regions.** We visualize the local landscape geometry evolution of training runs in Figure 2. For each model size, we select the training run with LR $2^{-10}$. **(Top).** Evolution of the top eigenvalues of the Hessian across iterations: $\lambda_i(\mathbf{H}(\boldsymbol{\theta}_t))$ vs. iteration $t$. **(Bottom).** One-dimensional loss landscape along a random perturbation direction: the perturbed loss $L(\boldsymbol{\theta}_t + \alpha\boldsymbol{\delta})$ vs. perturbation coefficient $\alpha$, shown across early training iterations $t$.

We next characterize the one-step loss change as $\eta$ approaches the stability boundary $2/\lambda_{\max}(\mathbf{S}_k)$.

**Lemma 4.2** (One-step Loss Change). *Let* $\boldsymbol{\delta}_k := \boldsymbol{\theta}_{k+1} - \boldsymbol{\theta}_k$. *Suppose that along the segment* $\{\boldsymbol{\theta}_k + \alpha\boldsymbol{\delta}_k : \alpha \in [0,1]\}$, *we have* $0 \leq \lambda_{\min}(\mathbf{S}(\boldsymbol{\theta}_k + \alpha\boldsymbol{\delta}_k)) \leq \lambda_{\max}(\mathbf{S}(\boldsymbol{\theta}_k + \alpha\boldsymbol{\delta}_k)) \leq \Lambda_k$. *Then,*

$$L(\boldsymbol{\theta}_{k+1}) - L(\boldsymbol{\theta}_k) \leq -\eta(1 - \frac{1}{2}\eta\Lambda_k)(\nabla L(\boldsymbol{\theta}_k))^{\top}\mathbf{M}\nabla L(\boldsymbol{\theta}_k).$$

*In particular, if* $\eta \uparrow 2/\Lambda_k$, *the guaranteed decrease per step* $\boxed{(L(\boldsymbol{\theta}_k) - L(\boldsymbol{\theta}_{k+1}))/\eta \to 0}$.

Lemma 4.2 states that when $\eta$ is close to $2/\Lambda_k$, each update yields only a marginal decrease in loss. Together with Lemma 4.1, it is clear that training near the stability boundary naturally leads to characteristic loss spikes and plateaus.

Importantly, the stability boundary is determined by the sharpness of the loss landscape. To further address **Q1-2**, we analyze how sharpness evolves during the early phase of pre-training.

**The Early Dynamics: From Sharp to Flat Landscapes.** We study how the local landscape geometry, particularly the sharpness, evolves for training runs in Figure 2. Specifically, we track the evolution of the top eigenvalues of the Hessian[4] $\mathbf{H}(\boldsymbol{\theta}_t)$ during early pre-training. For the early checkpoints $\boldsymbol{\theta}_t$, we also visualize the one-dimensional loss landscape along a random direction by plotting the function $\mathcal{L}(\alpha) := L(\boldsymbol{\theta}_t + \alpha\boldsymbol{\delta})$ with $\boldsymbol{\delta} \sim \mathcal{N}(0, \mathbf{I})$. Li et al. (2018) showed that such random-direction visualizations reliably capture intrinsic properties of the loss landscape properties, such as sharpness. To ensure fair comparison across iterations, we fix the same random vector $\boldsymbol{\delta}$ for all $\boldsymbol{\theta}_t$.

In Figure 3 (top), the largest eigenvalues of the Hessian $\mathbf{H}(\boldsymbol{\theta}_t)$ start at high values[5] and then decrease sharply, indicating a substantial reduction in curvature along the sharpest direction. Furthermore, in Figure 3 (bottom), the loss landscape along a random direction progressively widens as training proceeds, confirming that the model shifts from sharp to flat regions even in the most directions.

**A Tuning Recipe: Larger Peak LR, Longer Warmup.** We have seen that training stability depends on sharpness: when the landscape is steep, only a sufficiently small LR can keep updates stable; and pre-training initially traverses from sharp landscapes to flatter ones. Now let us return to **Q1** and **Q2**:

**A1.** *If the warmup phase is shortened, the LR rises too quickly while the model is still in sharp regions, leading to loss spikes and plateaus.*

---

[4]Following Cohen et al. (2021), we use the `Lanczos` algorithm to calculate top eigenvalues of Hessian.

[5]In fact, at initialization, sharpness is extremely low but rises sharply after the first update. The sharpness curves reported in Figure 3 therefore start from the first iteration.

**A2.** *As training progresses, the landscape becomes flatter and the same LR no longer threatens stability, which explains why instability is confined to the very beginning.*

Therefore, in practice, we need a sufficiently long warmup phase to keep the LR small until sharpness has decayed, thereby preventing loss spikes and plateaus. This rationale further suggests a practical tuning recipe: *the larger the peak LR, the longer the warmup should be*, ensuring iterates safely transition into flatter landscapes before reaching full step size.

To validate this, we train models with varied peak LRs $\eta$ and warmup lengths $T_w$ (in iterations). In Figure 4, within a LR range of $2^{-8}$ to $2^{-11}$, larger peak LRs require *proportionally* longer warmup to achieve the optimal validation loss $L(\boldsymbol{\theta}_{\text{best}})$. However, this proportionality does not hold universally. When $\eta = 2^{-7}$, the optimal warmup length remains $2^{10}$ iterations, the same as for $\eta = 2^{-8}$. Thus, the relationship applies within a reasonable range, when both the peak LR and warmup length are neither too small nor too large.

**Comparison with** Gilmer et al. (2022); Kalra & Barkeshli (2024). These works also studied warmup from a sharpness perspective but focused mainly on standard image classification tasks (e.g., ResNet on CIFAR-10) and full-batch gradient descent. In contrast, our work investigates warmup in the context of *large-scale* language model pre-training, visualizing the sharpness evolution under a general and practical training setup.

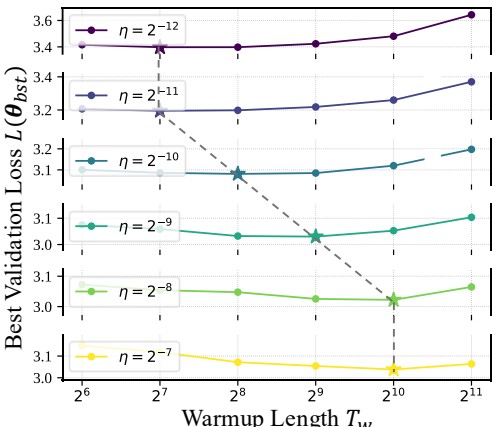

Figure 4: **Larger Peak LR, Longer Warmup.** We train a series of `LLaMA-2` models with 170M parameters and 100 TPP. We vary the peak LRs $\eta$ and warmup lengths $T_w$. We plot the best validation loss $L(\boldsymbol{\theta}_{\text{bst}})$ vs. $T_w$ for different $\eta$. For each $\eta$, the optimal $T_w$ is highlighted with a star.

## 5 LATE IN PRE-TRAINING: LOCAL LANDSCAPE GOVERNED BY NOISE SCALE

In this section, we turn to the local landscape geometry in the late phase. We observe that BS plays a central role: training with a large BS tends to find a *deeper* basin of the landscape, whereas a small BS favors a *wider* basin. Theoretically, we prove that this trade-off between *widen* or *deepen* is governed by the *noise scale*. Building on this, we propose a BS scheduler for the data-limited regime: *use small BS early and ramp the BS late*, which consumes fewer tokens to achieve the same loss.

**The Effect of BS: Local Landscapes Late in Training.** We conduct experiments to systematically investigate the role of BS in shaping the local landscape geometry during the late phase of pre-training. Specifically, we train models with different BSs for $T = 20{,}480$ iterations. Figure 5 (top left) shows the validation loss curves for each run. Evidently, larger BS consistently leads to lower terminal loss and faster convergence in term of iterations[6]. We then visualize the loss landscape around the final iterate $\boldsymbol{\theta}_T$. In Figure 5 (top right), it is clear that small BS produces flatter basins, whereas large BS yields deeper ones. To further demonstrate, Figure 5 (bottom) compares the landscape evolution of runs with $B = 0.49$M and $B = 7.8$M, indicating that in the late training phase, larger BS tends to deepen the basin, while smaller BS shifts toward wider basins.

Despite these results, two key questions remain:

**Q3.** *Why is there a trade-off between widening and deepening the basin?*

**Q4.** *Which factor underlying the hyperparameter BS governs this trade-off?*

To delve into **Q3-4**, we revisit the stochastic differential equation (SDE) in Jastrzębski et al. (2017).

**Widen or Deepen: Noise Scale Governs Basin Selection.** Following Jastrzębski et al. (2017), we take the continuous-time limit of Equation (1). Suppose that the noise covariance satisfies[7] $\frac{\eta}{B}\mathbf{M}\boldsymbol{\Sigma}(\boldsymbol{\theta}^\star)\mathbf{M}^\top = 2\tau\mathbf{M} + \mathcal{O}(\eta)$ for some temperature $\tau > 0$. As $\eta \to 0$, the scaled discrete process

---

[6]In terms of processed tokens, small BS training converges faster.

[7]The assumption of noise covariance is justified in Section C.2.

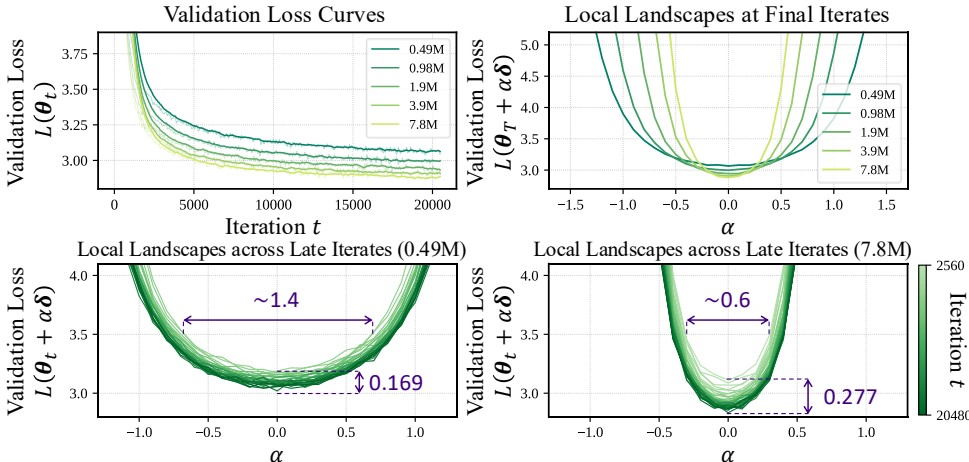

Figure 5: **Large BS deepens the basin, small BS widens the basin.** We train a series of `LLaMA-2` models (170M) for $T = 20{,}480$ iterations, using BSs $B \in \{0.49\text{M}, 0.98\text{M}, 1.9\text{M}, 3.9\text{M}, 7.8\text{M}\}$. **(Top left).** Validation loss curves for different BSs: $L(\boldsymbol{\theta}_t)$ vs. training iteration $t$. **(Top right).** One-dimensional loss landscapes at the final iterates $\boldsymbol{\theta}_T$ along a random perturbation direction: perturbed loss $L(\boldsymbol{\theta}_T + \alpha \boldsymbol{\delta})$ vs. perturbation coefficient $\alpha$, visualized across different BSs. **(Bottom).** One-dimensional loss landscape: the perturbed loss $L(\boldsymbol{\theta}_t + \alpha \boldsymbol{\delta})$ vs. perturbation coefficient $\alpha$, shown across late training iterations $t$ for $B = 0.49$M and $B = 7.8$M.

$\boldsymbol{\theta}_{\lfloor t/\eta \rfloor}$ converges weakly to the Itô SDE:

$$d\boldsymbol{\theta}_t = -\mathbf{M}\nabla L(\boldsymbol{\theta}_t)dt + \sqrt{2\tau}\mathbf{M}^{1/2}dW_t \tag{5}$$

where $W_t$ is standard Brownian motion and the noise scale $\tau$ is proportional to $\eta/B$.

Building on Equation (5) and the local quadratic model in Equation (3), we establish the trade-off between deepening and widening the loss basin.

**Theorem 5.1** (Depth-Flatness Trade-off). *Let the empirical risk $L(\boldsymbol{\theta})$ admit multiple local minima $\{\boldsymbol{\theta}_i^\star\}_{i=1}^m$ with Hessians $\mathbf{H}(\boldsymbol{\theta}_i^\star) \succ 0$. Under the SDE in Equation (5) with temperature $\tau$, the stationary probability that training resides in basin $i$ is given by:*

$$P_\tau(basin\ i) = \frac{\exp(-F_i(\tau)/\tau)}{\sum_j \exp(-F_j(\tau)/\tau)}, \quad F_i(\tau) := \boxed{L(\boldsymbol{\theta}_i^\star)} + \boxed{\frac{\tau}{2} \log \det \mathbf{H}(\boldsymbol{\theta}_i^\star)}.$$

Theorem 5.1 states that the basin selection is controlled by the free energy function $F(\tau) = L(\boldsymbol{\theta}^\star) + \frac{\tau}{2}\log\det\mathbf{H}(\boldsymbol{\theta}^\star)$. In early training, the loss term $L(\boldsymbol{\theta}^\star)$ dominates, so the model primarily seeks regions of lower loss. In later training, $L(\boldsymbol{\theta}^\star)$ is comparable to the flatness penalty $\log\det\mathbf{H}(\boldsymbol{\theta}^\star)$, and basin selection becomes increasingly sensitive to the noise scale $\tau \propto \eta/B$.

**Efficient Pre-Training: A BS Scheduler in Data-Limited Regime.** Turning back to **Q3** and **Q4**, the trade-off arises because basin selection balances loss minimization against curvature regularization (**A3**), with the governing factor being the noise scale $\tau$ (**A4**). Since the primary objective of pre-training is to minimize the training loss[8], this balance naturally favors largest BS available (small $\tau$). In practice, however, data availability is limited, and excessively large BS substantially increase data consumption[9]. Thus, scheduling BS in pre-training is crucial, particularly in the data-limited regime.

• **BS Scheduler: Design Principle I.** Inspired by our theory, loss reduction dominates early in training, during which large BS yields limited benefit. This suggests the first design principle.

> **Design Principle I.** *Start the training process with a small BS before increasing it later.*

Related ideas were noted by Li et al. (2025); Merrill et al. (2025), often referred to as *BS warmup*. However, a key difference in our design lies in when the BS should be increased. Surprisingly, we find that ramping BS *later in training* yields consistently *greater* performance.

---

[8]Note that our analysis focuses on how reduced noise (e.g., via larger BS) helps the optimizer move into deeper minima, conceptually different from the flat-minima perspective in fully interpolating regimes.

[9]For example, in Figure 5 (top right), when $B = 7.8$M, the run consumes approximately 160B tokens.

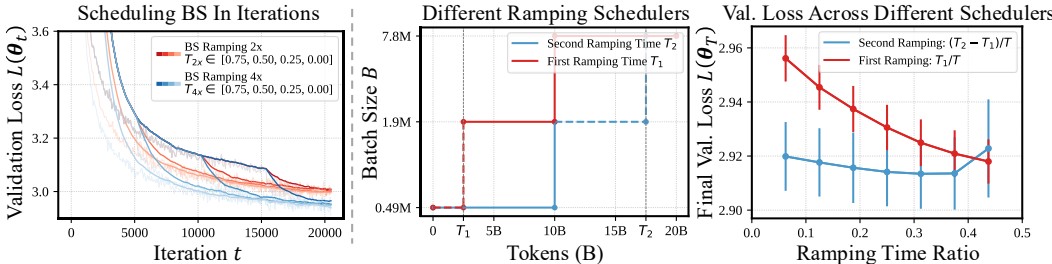

Figure 6: **(Left) Collapse of loss curves under different BS schedules.** Validation loss curves for training with different BS scheduling. In all runs, BS starts at 0.49M. For blue curves, BS is ramped up to $4\times$ its initial value; for red curves, BS is ramped to $2\times$. The ramping times, $T_{2\times}$ or $T_{4\times}$, are varied across different positions. **(Middle, Right) Ramping BS is more efficient late in training.** We evaluate a two-stage BS-ramping schedule with ramp times $T_1$ and $T_2$. For the red curves, we fix $T_2 = 10B$ and vary $T_1$; for the blue curves, we fix $T_1 = 10B$ and vary $T_2$. **(Middle).** Illustration of BS schedulers. **(Right).** Final validation loss vs. the relative ramping time, i.e., $(T_1)/T, (T_2 - T_1)/T \in [0, 0.5]$, where $T$ denotes the total training tokens.

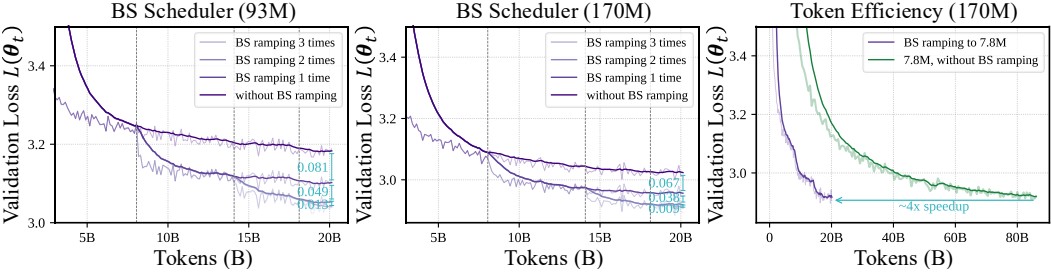

Figure 7: **BS scheduling improves data efficiency.** We train `LLaMA-2` models with 93M and 170M parameters, using a BS schedule that starts at 0.49M and increases by $4\times$ at each ramp. Models are trained with 1, 2, or 3 ramping steps, while models without ramping serve as the baseline. Vertical gray dashed lines indicate ramping positions. **(Left, Middle).** The validation curves for each run. **(Right).** Comparison between training with BS ramping to 7.8M and training with a fixed 7.8M BS.

• **BS Scheduler: Design Principle II.** To study this, we train models with different BS schedulers while keeping total training iterations fixed. In Figure 6 (left), all runs begin with an initial BS of 0.49M and ramp up to either $4\times$ or $2\times$ that value at different training iterations. Remarkably, all loss curves eventually collapse onto the same trajectory, regardless of when the BS ramping occurs. Note that when measured at the same training iteration, ramping the BS earlier results in higher data consumption. This indicates that early BS ramping offers no efficiency advantage, achieving the same loss but consuming more data.

We next evaluate BS schedules under a fixed token budget. Specifically, we consider a two-stage BS-ramping scheduler characterized by ramp times $T_1$ and $T_2$. To isolate the effect of each stage, we vary either $T_1$ or $T_2$ while keeping the other fixed. See Figure 6 (middle) for an illustration. In Figure 6 (right), a clear trend emerges: BS ramping is most effective when applied late in training (i.e., with $T_1$ and $T_2$ large), whereas ramping too early consistently harms final performance.

Together, since BS ramping ultimately leads all runs onto the same trajectory, delaying it allows maximal progress (lower loss) along that trajectory under a data-limited budget. This behavior also aligns with our theory. In the late phase of training, the flatness penalty becomes comparable to the loss term, and BS ramping sharply reduces the noise scale, driving rapid convergence toward deeper minima. This consistency between theory and practice leads to our second principle.

**Design Principle II.** *Ramp the batch size late in training—when loss reduction becomes marginal.*

To further validate our design principle, we train models using a BS schedule that starts at 0.49M and ramps by $4\times$ whenever loss minimization slows. In Figure 7 (left, middle), models with 1, 2 or 3 BS ramping steps achieves significant lower validation loss. While additional ramping steps provide diminishing returns, each step still offers a measurable improvement. Moreover, Figure 7 (right)

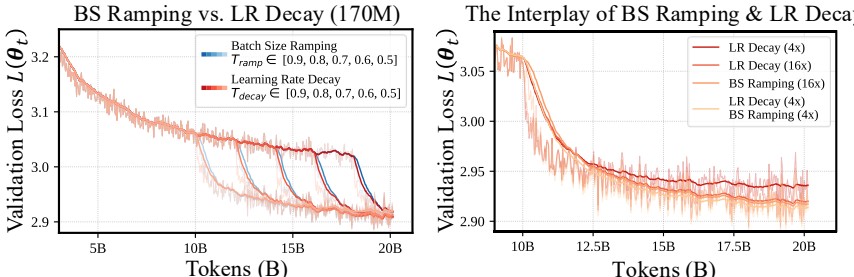

Figure 8: **(Left) BS ramping performs similarly to LR decay.** Validation loss curves for training with either BS ramping or LR decay. For BS ramping, BS increases to $16\times$ its initial value; for LR decay, the LR drops to $1/16$ of its initial value. Each method applies a single step at varying positions. **(Right) Interplay between BS ramping and LR decay.** We evaluate four different scheduling strategies. At the 10B tokens, (a) LR drops to $1/4$ of its initial value; (b) LR drops to $1/16$; (c) BS ramps to $16\times$. (d) LR drops to $1/4$ and BS ramps to $4\times$. In all runs (both left and right), BS starts at 0.49M and LR begins at $2^{-10}$ (after linear warmup).

highlights the data-efficiency of the BS scheduling: ramping the BS up to 7.8M achieves nearly the same final validation loss as training with a fixed 7.8M BS, but requires only about $\frac{1}{4}$ of the tokens (i.e., a $\sim 4\times$ speedup). These results confirm that our BS scheduling design preserves the benefits of large BS while substantially reducing data consumption.

**Comparison with** McCandlish et al. (2018); Merrill et al. (2025). McCandlish et al. (2018) linked BS scaling to the gradient noise and introduced the notion of CBS. Merrill et al. (2025) explored the BS scheduling (BS warmup), doubling BS once the CBS exceeds the current BS. We extend these works by showing that the noise scale governs the depth–flatness trade-off in basin selection, and by proposing a BS scheduling design principle that ramps the BS late in training.

## 6 MORE DISCUSSIONS: LR DECAY AND BS RAMPING

So far, we have excluded LR decay in our experiments to isolate the effect of BS ramping. Yet, recall that the noise scale $\tau$ is proportional to $\eta/B$. Our theory suggests that decaying the LR and ramping the BS both reduce the noise scale, and thus may have similar effects on basin selection.

**Comparing BS Ramping with LR Decay.** To study this, we train models using either BS ramping or LR decay. Both methods apply a one-time step change: BS ramping multiplies the BS by 16 at $T_{\text{ramp}}$, while LR decay divides the LR by 16 at $T_{\text{decay}}$. We align $T_{\text{ramp}}$ and $T_{\text{decay}}$ so that the changes occur at the same positions, enabling a direct comparison of their effects. In Figure 8 (left), BS ramping and LR decay produce remarkably similar validation loss curves across all change positions, consistent with the idea that both reduce the noise scale in comparable ways.

**Interacting BS Ramping with LR Decay.** Furthermore, we study the combined effect of using both BS ramping and LR decay. Specifically, we decay the LR by $4\times$ and simultaneously ramp the BS by $4\times$ at 10B tokens. We compare this hybrid schedule with three baselines: at the same point, we (a) drops the LR by $4\times$, (b) drops the LR by $16\times$ and (c) ramps the BS by $16\times$. We denote the hybrid schedule by (d). In Figure 8 (right), three of the schedules (b c d) produce nearly identical loss curves. Crucially, these three configurations yield the same noise scale, since they preserve the ratio $\eta/B$. In contrast, schedule (a) results in a noticeably different trajectory.

In summary, our results reinforce our theoretical prediction: training dynamics in the late phase are governed primarily by the noise scale $\tau \propto \eta/B$. LR decay reduce the noise scale in the same manner as BS ramping, and any hybrid scheduler that preserves $\eta/B$ will exhibit nearly identical behavior.

## 7 CONCLUSION AND LIMITATIONS

In conclusion, we present a unified theoretical and empirical view of how local landscape geometry evolves during language model pre-training. Our analysis reveals two phases: an early sharp-to-flat transition and a late noise-governed regime. The early dynamics explain the necessity of LR warmup, suggesting that larger peak LRs require proportionally longer warmup lengths. The late regime shows that noise scale controls a trade-off between widening and deepening loss basin, motivating a BS scheduling that starts with small BS and increases the BS late in training.

**Limitations.** The current theory primarily relies on strong assumptions, such as infinite-small LR in SDE. A natural future direction is to generalize the theory to more realistic settings. Additionally, the current theory cannot fully explain the collapse of loss curves under different BS schedules. Understanding the learning dynamics under different BS schedules remains an open question.

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

## A  TERMINOLOGIES

| Terminology | General Meaning | Usage in This Paper |
|---|---|---|
| Sharpness | A measure of curvature in the loss landscape, often characterized via the Hessian. Different works may define it differently. | We define sharpness as the curvature along the sharpest direction of the loss landscape. Mathematically, it is presented as the largest eigenvalue of the Hessian $\lambda_{\max}(\mathbf{H}(\boldsymbol{\theta}_t))$ or of the preconditioned curvature matrix $\lambda_{\max}(\mathbf{S}(\boldsymbol{\theta}_t))$. |
| Flat/sharp minimum | A minimum is a point where the gradient vanishes $\nabla L(\boldsymbol{\theta}) = 0$ and the loss does not decrease in a small neighborhood. A sharp minimum has large curvature; a flat minimum has small curvature. | We use these terms sparingly and follow the standard definitions from the sharpness/flat-minima literature. |
| Wide/deep basin | A loss basin is a region of the landscape surrounding a minimum. A wide basin rises loss slowly in most directions, whereas a deep basin has a significantly lower minimum value compared to its surroundings. | We use these terms to establish the depth–flatness trade-off: large noise scales tend to find wide basins, while small noise scales tend to find deeper regions with lower loss. |

Table 1: Terminology and usage in this paper.

## B  BROADER LIMITATIONS

While our work provides a unified geometric view of pre-training dynamics, it is subject to several broader limitations.

**Scale and data diversity.** Our experiments use decoder-only Transformers up to 530M parameters, trained on subsets of FineWeb-Edu with at most 1000 tokens-per-parameter. While this range is realistic for many pre-training settings, it is still much smaller than the multi-billion-parameter, multi-trillion-token regimes used in frontier models. The behavior of the early sharp-to-flat transition and the later noise-dominated regime may look different at those larger scales due to factors such as stronger path-dependence, data-mixture effects, or very long context windows. In addition, all of our experiments use English web text. Other domains—such as code, multilingual data, speech, or vision–language corpora, may exhibit different curvature patterns or gradient-noise characteristics. Further work is needed to understand how our observations generalize across scales and data types.

**Architectural coverage.** Our experiments focus mainly on LLaMA-style models that use RMSNorm and RoPE, along with a smaller set of GPT-2–like models that use LayerNorm, GELU, and different depth/width configurations. We observe the same qualitative early sharp-to-flat behavior across these families, but we do not systematically vary architectural components such as attention mechanisms, normalization placement (pre-norm vs. post-norm), activation functions, parameter sharing, or mixture-of-experts routing. We also do not yet study how architectural changes, such as replacing RMSNorm with LayerNorm or altering normalization statistics, affect the depth–flatness trade-off or the recommended learning-rate and batch-size schedules. A more comprehensive architectural study is left for future work.

**Optimizer and hyperparameter dependence.** Most of our experiments fix a particular optimizer configuration with standard hyperparameters, and gradient clipping. However, adaptive methods maintain evolving state (e.g., moving averages of first and second moments) whose transient behavior interacts non-trivially with curvature. Therefore, verifying our findings in a more general hyperparameter space is an important direction for future research. Additionally, we only considered the WSD-like schedulers. Other schedulers, such as `cosine` scheduler should be considered as well.

## C  MISSING PROOF

### C.1  EARLY IN PRE-TRAINING: LYAPUNOV STABILITY ANALYSIS

**Lemma C.1** (Stability Condition for Preconditioned GD). *Define the preconditioned curvature matrix $\mathbf{S}(\boldsymbol{\theta}_k) := \mathbf{M}^{1/2}\mathbf{H}(\boldsymbol{\theta}_k)\mathbf{M}^{1/2}$, and let $\{\lambda\}_{i=1}^p$ be the eigenvalues of $\mathbf{S}(\boldsymbol{\theta}_k)$. The linear system in Equation (4) is asymptotically stable (i.e., $\lim_{k\to\infty}\boldsymbol{e}_k = \mathbf{0}$) if $\eta$ satisfies*

$$0 < \eta < \frac{2}{\lambda_{\max}(\mathbf{S}(\boldsymbol{\theta}_k))}, \forall k \geq 0. \tag{6}$$

*Proof.* Since $\boldsymbol{e}_{k+1} = (\mathbf{I} - \eta\mathbf{M}\mathbf{H}_k)\boldsymbol{e}_k$, the linear system is asymptotically stable if all eigenvalues of $\mathbf{I} - \eta\mathbf{M}\mathbf{H}_k$ have magnitude less than 1. Note that:

$$\mathbf{I} - \eta\mathbf{M}\mathbf{H}_k = \mathbf{M}^{1/2}(\mathbf{I} - \eta\mathbf{S}_k)\mathbf{M}^{-1/2}, \tag{7}$$

so the eigenvalues are $1 - \eta\lambda_j(\mathbf{S}_k)$. The stability condition $|1 - \eta\lambda_j| < 1$ for all $j$ is equivalent to:

$$0 < \eta < \frac{2}{\lambda_{\max}(\mathbf{S}_k)}. \tag{8}$$

$\square$

**Lemma C.2** (Exact one-step loss change). *Define:*

$$\mathbf{S}(\boldsymbol{\theta}) := \mathbf{M}^{1/2}\mathbf{H}(\boldsymbol{\theta})\mathbf{M}^{1/2},$$

$$\boldsymbol{g}_k := \mathbf{M}^{1/2}\nabla L(\boldsymbol{\theta}_k),$$

$$\boldsymbol{\delta}_k := \boldsymbol{\theta}_{k+1} - \boldsymbol{\theta}_k = -\eta\mathbf{M}\nabla L(\boldsymbol{\theta}_k).$$

*Then the true loss change can be written exactly as*

$$L(\boldsymbol{\theta}_{k+1}) - L(\boldsymbol{\theta}_k) = -\eta\|\nabla L(\boldsymbol{\theta}_k)\|_{\mathbf{M}}^2 + \eta^2 \int_0^1 (1-t)\,\boldsymbol{g}_k^{\top}\mathbf{S}(\boldsymbol{\theta}_k + t\boldsymbol{\delta}_k)\boldsymbol{g}_k\,dt, \tag{9}$$

*where $\|\nabla L(\boldsymbol{\theta}_k)\|_{\mathbf{M}}^2 := (\nabla L(\boldsymbol{\theta}_k))^{\top}\mathbf{M}\nabla L(\boldsymbol{\theta}_k)$.*

*Proof.* Let $\boldsymbol{\delta}_k := \boldsymbol{\theta}_{k+1} - \boldsymbol{\theta}_k = -\eta\mathbf{M}\nabla L(\boldsymbol{\theta}_k)$ and define the scalar function

$$\phi(t) := L(\boldsymbol{\theta}_k + t\boldsymbol{\delta}_k), \quad t \in [0, 1].$$

Then

$$L(\boldsymbol{\theta}_{k+1}) - L(\boldsymbol{\theta}_k) = \phi(1) - \phi(0).$$

Compute the derivatives:

$$\phi'(t) = (\nabla L(\boldsymbol{\theta}_k + t\boldsymbol{\delta}_k))^{\top}\boldsymbol{\delta}_k,$$

$$\phi''(t) = \boldsymbol{\delta}_k^{\top}\mathbf{H}(\boldsymbol{\theta}_k + t\boldsymbol{\delta}_k)\boldsymbol{\delta}_k.$$

By Taylor's theorem with integral remainder:

$$\phi(1) - \phi(0) = \phi'(0) + \int_0^1 (1-t)\phi''(t)\,dt.$$

Now evaluate at $t = 0$:

$$\phi'(0) = (\nabla L(\boldsymbol{\theta}_k))^{\top}\boldsymbol{\delta}_k = -\eta(\nabla L(\boldsymbol{\theta}_k))^{\top}\mathbf{M}\nabla L(\boldsymbol{\theta}_k) = -\eta\|\nabla L(\boldsymbol{\theta}_k)\|_{\mathbf{M}}^2.$$

For the second derivative term:

$$\phi''(t) = \boldsymbol{\delta}_k^{\top}\mathbf{H}(\boldsymbol{\theta}_k + t\boldsymbol{\delta}_k)\boldsymbol{\delta}_k = \eta^2\boldsymbol{g}_k^{\top}\mathbf{S}(\boldsymbol{\theta}_k + t\boldsymbol{\delta}_k)\boldsymbol{g}_k,$$

since $\boldsymbol{\delta}_k = -\eta\mathbf{M}\nabla L(\boldsymbol{\theta}_k)$ and $\boldsymbol{g}_k = \mathbf{M}^{1/2}\nabla L(\boldsymbol{\theta}_k)$, and thus

$$\boldsymbol{\delta}_k^{\top}\mathbf{H}(\cdot)\boldsymbol{\delta}_k = \eta^2\boldsymbol{g}_k^{\top}\mathbf{S}(\cdot)\boldsymbol{g}_k.$$

Substituting both terms yields the result.

$\square$

**Lemma C.3** (One-step Loss Change). *Let $\boldsymbol{\delta}_k := \boldsymbol{\theta}_{k+1} - \boldsymbol{\theta}_k$. Suppose that along the segment $\{\boldsymbol{\theta}_k + \alpha\boldsymbol{\delta}_k : \alpha \in [0,1]\}$, we have $0 \leq \lambda_{\min}(\mathbf{S}(\boldsymbol{\theta}_k + \alpha\boldsymbol{\delta}_k)) \leq \lambda_{\max}(\mathbf{S}(\boldsymbol{\theta}_k + \alpha\boldsymbol{\delta}_k)) \leq \Lambda_k$. Then,*

$$L(\boldsymbol{\theta}_{k+1}) - L(\boldsymbol{\theta}_k) \leq -\eta(1 - \tfrac{1}{2}\eta\Lambda_k)(\nabla L(\boldsymbol{\theta}_k))^\top \mathbf{M} \nabla L(\boldsymbol{\theta}_k).$$

*In particular, if $\eta \leq 2/\Lambda_k$, each update is guaranteed to non-increasing in loss, i.e., $L(\boldsymbol{\theta}_{k+1}) \leq L(\boldsymbol{\theta}_k)$. Instead, if $\eta \uparrow 2/\Lambda_k$, the guaranteed decrease per step $\boxed{(L(\boldsymbol{\theta}_k) - L(\boldsymbol{\theta}_{k+1}))/\eta \to 0}$.*

*Proof.* From Lemma C.2, we have

$$\boldsymbol{g}_k^\top \mathbf{S}(\boldsymbol{\theta}_k + t\boldsymbol{\delta}_k)\boldsymbol{g}_k \leq \Lambda_k \|\boldsymbol{g}_k\|^2$$

for all $t$, since $\mathbf{S}(\cdot)$ is symmetric. Therefore,

$$L(\boldsymbol{\theta}_{k+1}) - L(\boldsymbol{\theta}_k) \leq -\eta\|\nabla L(\boldsymbol{\theta}_k)\|_{\mathbf{M}}^2 + \eta^2 \Lambda_k \|\boldsymbol{g}_k\|^2 \int_0^1 (1-t)\, dt$$

$$= -\eta\|\nabla L(\boldsymbol{\theta}_k)\|_{\mathbf{M}}^2 + \tfrac{1}{2}\eta^2 \Lambda_k \|\boldsymbol{g}_k\|^2.$$

Note that $\|\boldsymbol{g}_k\|^2 = \|\nabla L(\boldsymbol{\theta}_k)\|_{\mathbf{M}}^2$, yielding the result.

$\square$

## C.2 LATE IN PRE-TRAINING: SDE ANALYSIS

### C.2.1 DISCRETE-TIME SOLUTION

**Lemma C.4** (Eigenbasis Decomposition). *Let $\mathbf{S} := \mathbf{M}^{1/2}\mathbf{H}(\boldsymbol{\theta}^\star)\mathbf{M}^{1/2}$ with eigendecomposition $\mathbf{S} = \mathbf{Q}\boldsymbol{\Lambda}\mathbf{Q}^\top$, $\boldsymbol{\Lambda} = diag(\lambda_1, \ldots, \lambda_d)$. Define $\mathbf{G} := \mathbf{Q}^\top \mathbf{M}^{1/2}\boldsymbol{\Sigma}(\boldsymbol{\theta}^\star)\mathbf{M}^{1/2}\mathbf{Q}/B$. In coordinates $\boldsymbol{w}_k := \mathbf{Q}^\top\mathbf{M}^{-1/2}\boldsymbol{e}_k$, the recursion gives:*

$$\boldsymbol{w}_{k+1} = (\mathbf{I} - \eta\boldsymbol{\Lambda})\boldsymbol{w}_k + \eta\boldsymbol{\zeta}_k, \quad \mathbb{E}[\boldsymbol{\zeta}_k\boldsymbol{\zeta}_k^\top] = \mathbf{G}$$

*The stationary covariance $\boldsymbol{\Sigma}_w$ has diagonal elements:*

$$(\boldsymbol{\Sigma}_w)_{jj} = \frac{\eta^2 \mathbf{G}_{jj}}{1 - (1 - \eta\lambda_j)^2} = \frac{\eta \mathbf{G}_{jj}}{2\lambda_j - \eta\lambda_j^2} \tag{10}$$

*Proof.* First, we verify that $\mathbf{S} = \mathbf{M}^{1/2}\mathbf{H}(\boldsymbol{\theta}^\star)\mathbf{M}^{1/2}$ can be eigendecomposed. Since both $\mathbf{M}$ and $\mathbf{H}(\boldsymbol{\theta}^\star)$ are positive definite matrices, $\mathbf{S}$ is also positive definite matrix. By the spectral theorem, $\mathbf{S}$ admits the eigendecomposition $\mathbf{S} = \mathbf{Q}\boldsymbol{\Lambda}\mathbf{Q}^\top$, where $\mathbf{Q}$ is orthogonal and $\boldsymbol{\Lambda} = \text{diag}(\lambda_1, \ldots, \lambda_d)$ with $\lambda_i > 0$.

Starting from $\boldsymbol{e}_{k+1} = \mathbf{A}\boldsymbol{e}_k + \eta\mathbf{M}\boldsymbol{\xi}_k$ with $\mathbf{A} = \mathbf{I} - \eta\mathbf{M}\mathbf{H}(\boldsymbol{\theta}^\star)$, we change variables to $\boldsymbol{w}_k = \mathbf{Q}^\top\mathbf{M}^{-1/2}\boldsymbol{e}_k$.

$$\boldsymbol{w}_{k+1} = \mathbf{Q}^\top\mathbf{M}^{-1/2}\boldsymbol{e}_{k+1} = \mathbf{Q}^\top\mathbf{M}^{-1/2}(\mathbf{A}\boldsymbol{e}_k + \eta\mathbf{M}\boldsymbol{\xi}_k)$$

$$= \mathbf{Q}^\top\mathbf{M}^{-1/2}(\mathbf{I} - \eta\mathbf{M}\mathbf{H}(\boldsymbol{\theta}^\star))\boldsymbol{e}_k + \eta\mathbf{Q}^\top\mathbf{M}^{1/2}\boldsymbol{\xi}_k$$

$$= \mathbf{Q}^\top\mathbf{M}^{-1/2}\boldsymbol{e}_k - \eta\mathbf{Q}^\top\mathbf{M}^{1/2}\mathbf{H}(\boldsymbol{\theta}^\star)\boldsymbol{e}_k + \eta\mathbf{Q}^\top\mathbf{M}^{1/2}\boldsymbol{\xi}_k$$

$$= \boldsymbol{w}_k - \eta\mathbf{Q}^\top\mathbf{M}^{1/2}\mathbf{H}(\boldsymbol{\theta}^\star)\mathbf{M}^{1/2}\mathbf{Q}\boldsymbol{w}_k + \eta\mathbf{Q}^\top\mathbf{M}^{1/2}\boldsymbol{\xi}_k$$

$$= \boldsymbol{w}_k - \eta\mathbf{Q}^\top\mathbf{S}\mathbf{Q}\boldsymbol{w}_k + \eta\mathbf{Q}^\top\mathbf{M}^{1/2}\boldsymbol{\xi}_k$$

$$= (\mathbf{I} - \eta\boldsymbol{\Lambda})\boldsymbol{w}_k + \eta\mathbf{Q}^\top\mathbf{M}^{1/2}\boldsymbol{\xi}_k$$

Defining $\boldsymbol{\zeta}_k := \mathbf{Q}^\top\mathbf{M}^{1/2}\boldsymbol{\xi}_k$, we get:

$$\mathbb{E}[\boldsymbol{\zeta}_k\boldsymbol{\zeta}_k^\top] = \mathbf{Q}^\top\mathbf{M}^{1/2}\mathbb{E}[\boldsymbol{\xi}_k\boldsymbol{\xi}_k^\top]\mathbf{M}^{1/2}\mathbf{Q} = \mathbf{Q}^\top\mathbf{M}^{1/2}\boldsymbol{\Sigma}(\boldsymbol{\theta}^\star)\mathbf{M}^{1/2}\mathbf{Q}/B =: \mathbf{G}$$

As matrix $(\mathbf{I} - \eta\mathbf{\Lambda})$ is diagonal, the recursion now decouples into independent scalar equations for each component $j$:

$$(\boldsymbol{w}_{k+1})_j = (1 - \eta\lambda_j)(\boldsymbol{w}_k)_j + \eta(\boldsymbol{\zeta}_k)_j.$$

For each component $j$, the stationary variance satisfies:

$$(\mathbf{\Sigma}_w)_{jj} = (1 - \eta\lambda_j)^2(\mathbf{\Sigma}_w)_{jj} + \eta^2\mathbf{G}_{jj} \tag{11}$$

Solving for $(\mathbf{\Sigma}_w)_{jj}$:

$$(\mathbf{\Sigma}_w)_{jj} = \frac{\eta^2\mathbf{G}_{jj}}{1 - (1 - \eta\lambda_j)^2} = \frac{\eta^2\mathbf{G}_{jj}}{1 - (1 - 2\eta\lambda_j + \eta^2\lambda_j^2)}$$

$$= \frac{\eta^2\mathbf{G}_{jj}}{2\eta\lambda_j - \eta^2\lambda_j^2} = \frac{\eta\mathbf{G}_{jj}}{2\lambda_j - \eta\lambda_j^2}$$

$\square$

### C.2.2 CONTINUOUS-TIME LIMIT

We now take the continuous-time limit ($\eta \to 0$) to derive a simpler universal theory. The exact solution for the variance in the eigenbasis from Lemma C.4, i.e., $(\mathbf{\Sigma}_w)_{jj} = \eta\mathbf{G}_{jj}/(2\lambda_j - \eta\lambda_j^2)$, guides the necessary scaling for the continuous-time limit. Because $(\mathbf{\Sigma}_w)_{jj}$ converges to a finite non-zero value as $\eta \to 0$, the numerator $\eta\mathbf{G}_{jj}$ must remain finite. This suggests defining a quantity $\tau$ such that for each mode $j$:

$$\eta\,\mathbf{G}_{jj} \to 2\tau \quad \text{as} \quad \eta \to 0.$$

We strengthen this to :

$$\eta\,\mathbf{G} \to 2\tau\mathbf{I} \quad \text{as} \quad \eta \to 0.$$

Recalling that $\mathbf{G} = \mathbf{Q}^\top\mathbf{M}^{1/2}\mathbf{\Sigma}(\boldsymbol{\theta}^\star)\mathbf{M}^{1/2}\mathbf{Q}/B$, this condition in the original coordinate system translates to the required scaling for the noise covariance:

$$\frac{\eta}{B}\mathbf{M}\mathbf{\Sigma}(\boldsymbol{\theta}^\star)\mathbf{M}^\top \to 2\tau\mathbf{M}.$$

**Proposition C.1** (Convergence to SDE). *Consider the scaled discrete process $\boldsymbol{\theta}_{\lfloor t/\eta \rfloor}$ as $\eta \to 0$. Suppose the noise covariance satisfies*

$$\frac{\eta}{B}\mathbf{M}\mathbf{\Sigma}(\boldsymbol{\theta}^\star)\mathbf{M}^\top = 2\tau\mathbf{M} + O(\eta), \tag{12}$$

*for some temperature $\tau > 0$. Then the process converges weakly to the Itô SDE:*

$$d\boldsymbol{\theta}_t = -\mathbf{M}\nabla L(\boldsymbol{\theta}_t)dt + \sqrt{2\tau}\mathbf{M}^{1/2}dW_t \tag{13}$$

*where $W_t$ is standard Brownian motion.*

*Proof.* Consider the discrete preconditioned SGD update:

$$\boldsymbol{\theta}_{k+1} = \boldsymbol{\theta}_k - \eta\mathbf{M}(\nabla L(\boldsymbol{\theta}_k) + \boldsymbol{\xi}_k),$$

Define the scaled process $\boldsymbol{\theta}^{(\eta)}(t) = \boldsymbol{\theta}_{\lfloor t/\eta \rfloor}$. The increment $\Delta\boldsymbol{\theta}_k = \boldsymbol{\theta}_{k+1} - \boldsymbol{\theta}_k$ satisfies:

$$\mathbb{E}[\Delta\boldsymbol{\theta}_k \mid \boldsymbol{\theta}_k = \boldsymbol{\theta}] = -\eta\mathbf{M}\nabla L(\boldsymbol{\theta}),$$

$$\mathrm{Cov}(\Delta\boldsymbol{\theta}_k \mid \boldsymbol{\theta}_k = \boldsymbol{\theta}) = \frac{\eta^2}{B}\mathbf{M}\mathbf{\Sigma}(\boldsymbol{\theta}^\star)\mathbf{M}^\top.$$

Given the scaling condition Equation (12), the covariance is $O(\eta)$.

The generator $\mathcal{L}^{(\eta)}$ of the discrete process for a smooth function $\boldsymbol{f}$ is:

$$\mathcal{L}^{(\eta)}\boldsymbol{f}(\boldsymbol{\theta}) = \frac{1}{\eta}\mathbb{E}[\boldsymbol{f}(\boldsymbol{\theta}_{k+1}) - \boldsymbol{f}(\boldsymbol{\theta}_k) \mid \boldsymbol{\theta}_k = \boldsymbol{\theta}].$$

Using a Taylor expansion and taking conditional expectation:

$$\mathbb{E}[\boldsymbol{f}(\boldsymbol{\theta}_{k+1}) - \boldsymbol{f}(\boldsymbol{\theta}_k) \mid \boldsymbol{\theta}] = -\eta \nabla \boldsymbol{f}(\boldsymbol{\theta})^\top \mathbf{M} \nabla L(\boldsymbol{\theta}) + \frac{1}{2}\mathbb{E}[(\Delta\boldsymbol{\theta})^\top \nabla^2 \boldsymbol{f}(\boldsymbol{\theta})\Delta\boldsymbol{\theta}] + O(\eta^{3/2}).$$

For the second term, with $\Delta\boldsymbol{\theta} = -\eta M(\nabla L(\boldsymbol{\theta}) + \boldsymbol{\xi}_k)$:

$$
\begin{aligned}
\mathbb{E}[(\Delta\boldsymbol{\theta})^\top \nabla^2 \boldsymbol{f}(\boldsymbol{\theta})\Delta\boldsymbol{\theta}] &= \eta^2 \mathbb{E}[(\nabla L(\boldsymbol{\theta}) + \boldsymbol{\xi}_k)^\top \mathbf{M}^\top \nabla^2 \boldsymbol{f}(\boldsymbol{\theta})\mathbf{M}(\nabla L(\boldsymbol{\theta}) + \boldsymbol{\xi}_k)] \\
&= \eta^2 \mathbb{E}[\boldsymbol{\xi}_k^\top \mathbf{M}^\top \nabla^2 \boldsymbol{f}(\boldsymbol{\theta})\mathbf{M}\boldsymbol{\xi}_k] + O(\eta^2) \\
&= \eta^2 \mathrm{Tr}(\mathbf{M}^\top \nabla^2 \boldsymbol{f}(\boldsymbol{\theta})\mathbf{M}\mathbb{E}[\boldsymbol{\xi}_k\boldsymbol{\xi}_k^\top]) + O(\eta^2) \\
&= \frac{\eta^2}{B}\mathrm{Tr}(\mathbf{M}^\top \nabla^2 \boldsymbol{f}(\boldsymbol{\theta})\mathbf{M}\boldsymbol{\Sigma}(\boldsymbol{\theta}^\star)) + O(\eta^2) \\
&= \frac{\eta^2}{B}\mathrm{Tr}(\mathbf{M}\boldsymbol{\Sigma}(\boldsymbol{\theta}^\star)\mathbf{M}^\top \nabla^2 \boldsymbol{f}(\boldsymbol{\theta})) + O(\eta^2)
\end{aligned}
$$

where we used $\mathbb{E}[\boldsymbol{\xi}^\top \mathbf{A}\boldsymbol{\xi}] = \mathrm{Tr}(A\mathbb{E}[\boldsymbol{\xi}\boldsymbol{\xi}^\top])$ and trace cyclicity $\mathrm{Tr}(\mathbf{ABC}) = \mathrm{Tr}(\mathbf{CAB})$.

Therefore:

$$\frac{1}{2}\mathbb{E}[(\Delta\boldsymbol{\theta})^\top \nabla^2 \boldsymbol{f}(\boldsymbol{\theta})\Delta\boldsymbol{\theta}] = \frac{\eta^2}{2B}\mathrm{Tr}(\mathbf{M}\boldsymbol{\Sigma}(\boldsymbol{\theta}^\star)\mathbf{M}^\top \nabla^2 \boldsymbol{f}(\boldsymbol{\theta})) + O(\eta^2).$$

Using the scaling condition Equation (12), we have :

$$\frac{\eta^2}{2B}\mathrm{Tr}(\mathbf{M}\boldsymbol{\Sigma}(\boldsymbol{\theta}^\star)\mathbf{M}^\top \nabla^2 \boldsymbol{f}(\boldsymbol{\theta})) = \frac{\eta}{2}\mathrm{Tr}\left(2\tau\mathbf{M}\nabla^2 \boldsymbol{f}(\boldsymbol{\theta})\right) + O(\eta^2) = \eta\tau\mathrm{Tr}(\mathbf{M}\nabla^2 \boldsymbol{f}(\boldsymbol{\theta})) + O(\eta^2).$$

Thus,

$$\mathcal{L}^{(\eta)}\boldsymbol{f}(\boldsymbol{\theta}) = -\nabla \boldsymbol{f}(\boldsymbol{\theta})^\top \mathbf{M}\nabla L(\boldsymbol{\theta}) + \tau\mathrm{Tr}(\mathbf{M}\nabla^2 \boldsymbol{f}(\boldsymbol{\theta})) + O(\eta).$$

As $\eta \to 0$, $\mathcal{L}^{(\eta)}\boldsymbol{f}(\boldsymbol{\theta})$ converges to:

$$\mathcal{L}\boldsymbol{f}(\boldsymbol{\theta}) = -\nabla \boldsymbol{f}(\boldsymbol{\theta})^\top \mathbf{M}\nabla L(\boldsymbol{\theta}) + \tau\mathrm{Tr}(\mathbf{M}\nabla^2 \boldsymbol{f}(\boldsymbol{\theta})),$$

which is the generator of the Itô SDE:

$$d\boldsymbol{\theta}_t = -\mathbf{M}\nabla L(\boldsymbol{\theta}_t)dt + \sqrt{2\tau}\mathbf{M}^{1/2}dW_t.$$

By the weak convergence theory (e.g., via the martingale problem or generator convergence), the process $\boldsymbol{\theta}^{(\eta)}(t)$ converges weakly to the solution of this SDE. $\qquad\square$

**Proposition C.2** (Gibbs Stationary Distribution). *The SDE in Equation* (13) *has stationary distribution:*

$$p_\infty(\boldsymbol{\theta}) \propto \exp(-L(\boldsymbol{\theta})/\tau) \tag{14}$$

*Proof.* The generator of the SDE (5) is $\mathcal{L}f = -\mathbf{M}\nabla L \cdot \nabla f + \tau\mathrm{tr}(\mathbf{M}\nabla^2 f)$. The Fokker-Planck equation for the probability density $p(t, \boldsymbol{\theta})$ is:

$$\partial_t p = \mathcal{L}^* p = \nabla \cdot (\mathbf{M}\nabla L\, p) + \tau\nabla \cdot (\mathbf{M}\nabla p)$$

where $\mathcal{L}^*$ is the adjoint operator. Setting $\partial_t p = 0$ for stationarity:

$$
\begin{aligned}
0 &= \nabla \cdot (\mathbf{M}\nabla L\, p_\infty) + \tau\nabla \cdot (\mathbf{M}\nabla p_\infty) \\
&= \nabla \cdot (\mathbf{M}\nabla L\, p_\infty + \tau\mathbf{M}\nabla p_\infty)
\end{aligned}
$$

This implies the current $J = \mathbf{M}\nabla L\, p_\infty + \tau\mathbf{M}\nabla p_\infty$ has zero divergence. For a potential-driven system, we require $J = \mathbf{0}$:

$$
\begin{aligned}
\mathbf{M}\nabla L\, p_\infty + \tau\mathbf{M}\nabla p_\infty &= \mathbf{0} \\
\nabla L\, p_\infty + \tau\nabla p_\infty &= \mathbf{0} \quad (\text{since } \mathbf{M} \succ 0) \\
\frac{\nabla p_\infty}{p_\infty} &= -\frac{\nabla L}{\tau}
\end{aligned}
$$

Integrating: $\log p_\infty = -L/\tau + \text{const}$, which gives Equation (14). $\qquad\square$

**Theorem C.1** (Free Energy Minimization). *Let the empirical risk $L(\boldsymbol{\theta})$ admit multiple local minima $\{\boldsymbol{\theta}_i^\star\}_{i=1}^m$ with Hessians $\mathbf{H}(\boldsymbol{\theta}_i^\star) \succ 0$. Under the SDE in Equation (13) with temperature $\tau$, the stationary probability that training resides in basin $i$ is given by:*

$$P_\tau(\text{basin } i) = \frac{\exp(-F_i(\tau)/\tau)}{\sum_j \exp(-F_j(\tau)/\tau)}, \quad F_i(\tau) := L(\boldsymbol{\theta}_i^\star) + \frac{\tau}{2} \log \det \mathbf{H}(\boldsymbol{\theta}_i^\star). \tag{15}$$

*Proof.* From the Gibbs distribution Equation (14), the probability mass in basin $i$ is:

$$P_\tau(\text{basin } i) = \frac{\int_{B_i} e^{-L(\boldsymbol{\theta})/\tau} d\boldsymbol{\theta}}{\int_{\mathbb{R}^d} e^{-L(\boldsymbol{\theta})/\tau} d\boldsymbol{\theta}}$$

where $B_i$ is the basin of attraction around minimum $\boldsymbol{\theta}_i^\star$.

For the numerator, using the quadratic approximation $L(\boldsymbol{\theta}) = L(\boldsymbol{\theta}_i^\star) + \frac{1}{2}(\boldsymbol{\theta} - \boldsymbol{\theta}_i^\star)^\top \mathbf{H}(\boldsymbol{\theta}_i^\star)(\boldsymbol{\theta} - \boldsymbol{\theta}_i^\star)$ in basin $i$ we get:

$$\int_{B_i} e^{-L(\boldsymbol{\theta})/\tau} d\boldsymbol{\theta} = \int_{\mathbb{R}^d} \exp\left(-\frac{L(\boldsymbol{\theta}_i^\star)}{\tau} - \frac{1}{2\tau}(\boldsymbol{\theta} - \boldsymbol{\theta}_i^\star)^\top \mathbf{H}(\boldsymbol{\theta}_i^\star)(\boldsymbol{\theta} - \boldsymbol{\theta}_i^\star)\right) d\boldsymbol{\theta}$$

$$= e^{-L(\boldsymbol{\theta}_i^\star)/\tau} \int_{\mathbb{R}^d} \exp\left(-\frac{1}{2\tau}(\boldsymbol{\theta} - \boldsymbol{\theta}_i^\star)^\top \mathbf{H}(\boldsymbol{\theta}_i^\star)(\boldsymbol{\theta} - \boldsymbol{\theta}_i^\star)\right) d\boldsymbol{\theta}$$

The integral is a multivariate Gaussian with covariance $\tau\mathbf{H}(\boldsymbol{\theta}_i^\star)^{-1}$. Using the standard formula for Gaussian integrals:

$$\int_{\mathbb{R}^d} \exp\left(-\frac{1}{2} y^\top \boldsymbol{\Sigma}^{-1} y\right) dy = (2\pi)^{d/2} (\det \boldsymbol{\Sigma})^{1/2}$$

With $\boldsymbol{\Sigma} = \tau\mathbf{H}(\boldsymbol{\theta}_i^\star)^{-1}$, we have $\det \boldsymbol{\Sigma} = \tau^d (\det \mathbf{H}(\boldsymbol{\theta}_i^\star))^{-1}$ and $\boldsymbol{\Sigma}^{-1} = \tau^{-1}\mathbf{H}(\boldsymbol{\theta}_i^\star)$:

$$\int_{\mathbb{R}^d} \exp\left(-\frac{1}{2\tau}(\boldsymbol{\theta} - \boldsymbol{\theta}_i^\star)^\top \mathbf{H}(\boldsymbol{\theta}_i^\star)(\boldsymbol{\theta} - \boldsymbol{\theta}_i^\star)\right) d\boldsymbol{\theta} = (2\pi)^{d/2} (\tau^d (\det \mathbf{H}(\boldsymbol{\theta}_i^\star))^{-1})^{1/2}$$

$$= (2\pi\tau)^{d/2} (\det \mathbf{H}(\boldsymbol{\theta}_i^\star))^{-1/2}$$

Therefore:

$$\int_{B_i} e^{-L(\boldsymbol{\theta})/\tau} d\boldsymbol{\theta} = e^{-L(\boldsymbol{\theta}_i^\star)/\tau} (2\pi\tau)^{d/2} (\det \mathbf{H}(\boldsymbol{\theta}_i^\star))^{-1/2}$$

$$= (2\pi\tau)^{d/2} \exp\left(-L(\boldsymbol{\theta}_i^\star)/\tau - \frac{1}{2} \log \det \mathbf{H}(\boldsymbol{\theta}_i^\star)\right)$$

$$= (2\pi\tau)^{d/2} \exp\left(-\frac{1}{\tau}\left(L(\boldsymbol{\theta}_i^\star) + \frac{\tau}{2} \log \det \mathbf{H}(\boldsymbol{\theta}_i^\star)\right)\right)$$

$$= (2\pi\tau)^{d/2} \exp(-F_i(\tau)/\tau)$$

Similarly, the total partition function is:

$$Z(\tau) = \int_{\mathbb{R}^d} e^{-L(\boldsymbol{\theta})/\tau} d\boldsymbol{\theta} = d\sum_{j=1}^m \int_{B_j} e^{-L(\boldsymbol{\theta})/\tau} d\boldsymbol{\theta}$$

$$= (2\pi\tau)^{d/2} \sum_{j=1}^m \exp(-F_j(\tau)/\tau)$$

Therefore:

$$P_\tau(\text{basin } i) = \frac{(2\pi\tau)^{d/2} \exp(-F_i(\tau)/\tau)}{(2\pi\tau)^{d/2} \sum_j \exp(-F_j(\tau)/\tau)} = \frac{\exp(-F_i(\tau)/\tau)}{\sum_j \exp(-F_j(\tau)/\tau)}$$

This completes the proof of the free energy formula Equation (15). $\qquad\square$

# D    EXPERIMENTAL SETUPS AND MORE RESULTS

## D.1    EXPERIMENTAL SETUPS

**Models.** We utilize two popular classes of LLM models for our pre-training experiments:

- **GPT-2.** We use GPT-2 (small) model (Radford et al., 2019), implemented via the nanoGPT code base (Karpathy, 2022). Following nanoGPT, the model employs Gaussian Error Linear Unit (GELU) activations and standard Layer Normalization (LayerNorm). Detailed model configurations are provided in Table 2.
- **LLaMA.** LLaMA (Touvron et al., 2023) is another popular decoder-only Transformer architecture, incorporating Rotary Positional Encoding (RoPE) (Su et al., 2024), Swish-Gated Linear Unit (SwiGLU), and Root mean square layer normalization (RMSNorm). For implementation, we utilize the LLaMA code from HuggingFace Transformers Library (Wolf et al., 2020). Additional model configurations are detailed in Table 2.

**Datasets.** Training is performed on the `FineWeb-Edu` dataset (Penedo et al., 2024). We adopt the a subset randomly sampled from the whole dataset of around 100B `GPT-2` tokens. The same dataset has been widely used in literature on LLM pre-training.

**Optimizers.** To generalize our findings across different optimizers, we choose:

- **AdamW.** AdamW (Kingma & Ba, 2014) is adopted with hyperparameters $\beta_1 = 0.95$, $\beta_2 = 0.95$, and weight decay $0.1$.
- **Muon.** Muon (Keller et al., 2024) is used with momentum of $0.95$ and weight decay $0.1$.
- **Adam-mini.** The hyperparameter of Adam-mini (Zhang et al., 2024b) is the same as AdamW.
- **Lion.** Lion (Chen et al., 2024) is used with hyperparameters $\beta_1 = 0.95$, $\beta_2 = 0.98$. The LR of Lion $\eta$ is divided by $10\times$ compared with the LR of AdamW in the same experiments, and the weight decay $\lambda$ is ramped up to $10\times$ to keep the effective LR $\lambda\eta = 0.1$.

All these optimizers are used with gradient clipping at $1.0$ for stability.

Table 2: Model configurations.

| Acronym | Size | $d_{\mathrm{model}}$ | $d_{\mathrm{FF}}$ | n_head | depth |
|---|---|---|---|---|---|
| GPT-2 (small) | 124M | 768 | 3072 | 12 | 12 |
| LLaMA (93M) | 93M | 512 | 2048 | 16 | 8 |
| LLaMA (170M) | 170M | 768 | 3072 | 12 | 8 |
| LLaMA (270M) | 270M | 1024 | 4096 | 16 | 8 |
| LLaMA (530M) | 530M | 1536 | 6144 | 24 | 8 |

## D.2    MORE RESULTS UNDER VARIOUS SETUPS.

In this section, we extend our findings to other architectures, optimization algorithms, and larger training scales. Due to computational constraints, we primarily focus on validating the sharp-to-flat early dynamics and the proposed BS scheduling principle across these settings. We also explore the warmup–tuning recipe on additional architectures.

**Extension to `GPT-2` Architectures.** See Figures 9 and 10 for details.

**Extension to Other Optimizers.** See Figure 11 for Adam-mini, and see Figure 12 for Lion.

**Extension to Larger Models.** See Figure 13 for models with 270M and 530M parameters.

## D.3    ABLATION STUDIES ON EARLY INSTABILITIES.

In this section, we conduct ablation studies on the root cause of instabilities, such as loss spikes and plateaus, observed in early training.

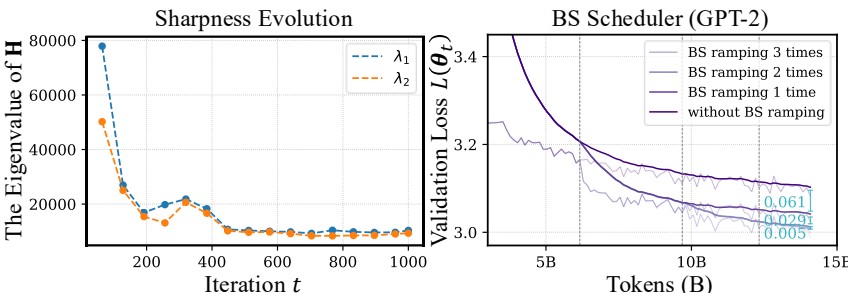

Figure 9: **Extensive to `GPT-2` architectures. (Left). Early sharp-to-flat dynamics.** Evolution of the top eigenvalues of the Hessian across iterations: $\lambda_i(\mathbf{H}(\boldsymbol{\theta}_t))$ vs. iteration $t$. **(Right). BS scheduling improves data efficiency. BS scheduling improves data efficiency.** We use a BS schedule that starts at $0.49$M and increases by $4\times$ at each ramp. Models are trained with $1, 2$, or $3$ ramping steps, while models without ramping serve as the baseline. Vertical gray dashed lines indicate ramping positions.

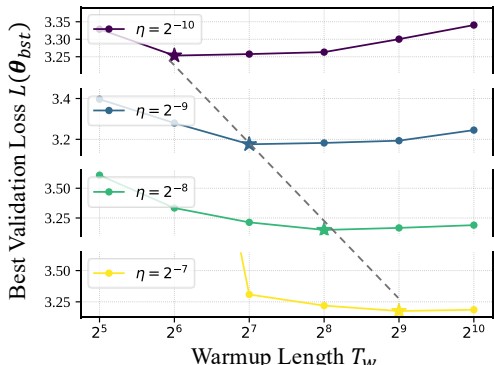

Figure 10: **Extensive to `GPT-2` architectures. Larger Peak LR, Longer Warmup.** We train a series of `GPT-2` models with 100 TPP. We vary the peak LRs $\eta$ and warmup lengths $T_w$. We plot the best validation loss $L(\boldsymbol{\theta}_{\text{bst}})$ vs. $T_w$ for different $\eta$. The optimal $T_w$ is highlighted with a star.

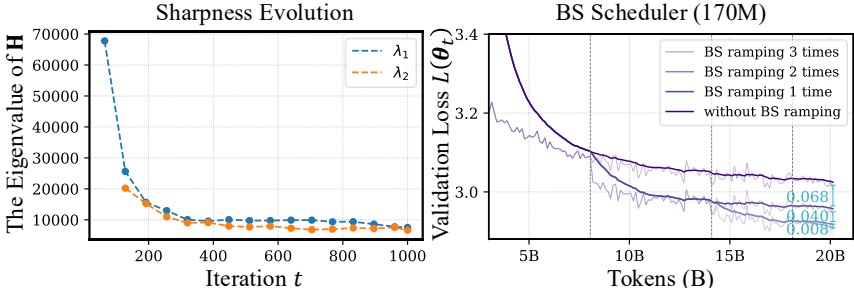

Figure 11: **Extensive to Adam-mini optimizer. (Left). Early sharp-to-flat dynamics.** Evolution of the top eigenvalues of the Hessian across iterations: $\lambda_i(\mathbf{H}(\boldsymbol{\theta}_t))$ vs. iteration $t$. **(Right). BS scheduling improves data efficiency. BS scheduling improves data efficiency.** We use a BS schedule that starts at $0.49$M and increases by $4\times$ at each ramp. Models are trained with $1, 2$, or $3$ ramping steps, while models without ramping serve as the baseline. Vertical gray dashed lines indicate ramping positions.

**Is it the unstable optimizer?** To disentangle optimizer-induced instability from landscape-induced instability, we repeated the experiments using Muon, a substantially more stable optimizer than AdamW. In Figure 14, the loss spikes and plateaus consistently occurs under Muon when warmup is shortened or the peak LR is increased. This rules out the possibility that the behavior stems from AdamW's startup issues.

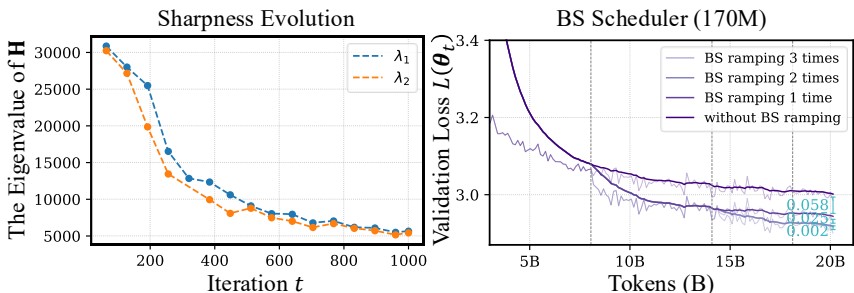

Figure 12: **Extensive to Lion optimizer. (Left). Early sharp-to-flat dynamics.** Evolution of the top eigenvalues of the Hessian across iterations: $\lambda_i(\mathbf{H}(\boldsymbol{\theta}_t))$ vs. iteration $t$. **(Right). BS scheduling improves data efficiency. BS scheduling improves data efficiency.** We use a BS schedule that starts at $0.49$M and increases by $4\times$ at each ramp. Models are trained with $1, 2$, or $3$ ramping steps, while models without ramping serve as the baseline. Vertical gray dashed lines indicate ramping positions.

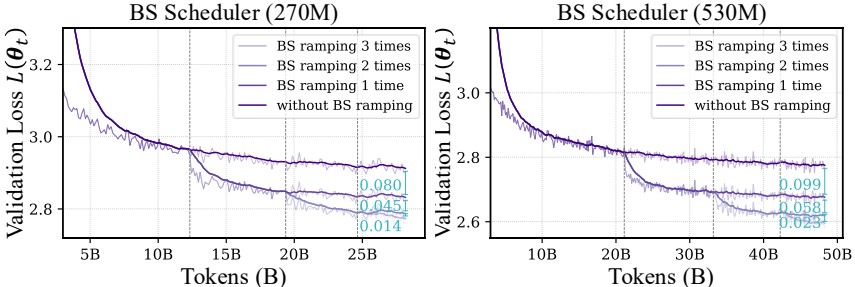

Figure 13: **Extensive to Larger Scale. (Left). Early sharp-to-flat dynamics.** Evolution of the top eigenvalues of the Hessian across iterations: $\lambda_i(\mathbf{H}(\boldsymbol{\theta}_t))$ vs. iteration $t$. **(Right). BS scheduling improves data efficiency. BS scheduling improves data efficiency.** We use a BS schedule that starts at $0.49$M and increases by $4\times$ at each ramp. Models are trained with $1, 2$, or $3$ ramping steps, while models without ramping serve as the baseline. Vertical gray dashed lines indicate ramping positions.

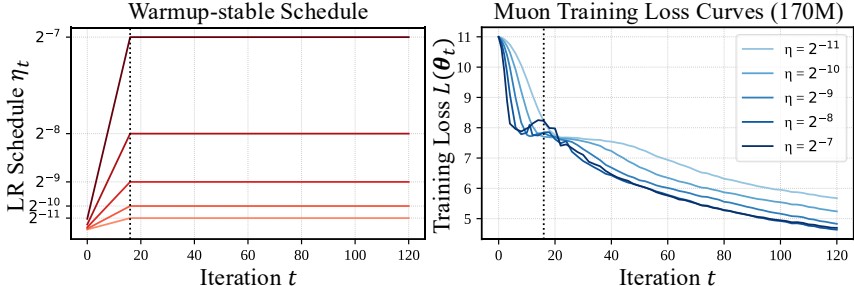

Figure 14: **Muon consistently shows loss spikes and plateaus early in training.** We train a series of `LLaMA-2` models with 170M parameters. We adopt a warmup-stable schedule, where the warmup length is shortened to 16 iterations and the peak LR is varied, $\eta \in \{2^{-11}, 2^{-10}, 2^{-9}, 2^{-8}, 2^{-7}\}$. **(Left).** LR schedule: $\eta_t$ vs. training iteration $t$. **(Middle, Right).** Training loss curves for different model sizes: $L(\boldsymbol{\theta}_t)$ vs. training iteration $t$. The vertical dashed line marks the end of the warmup phase.

**What if we use longer warmup?** We vary only the warmup length while fixing the peak LR at $2^{-7}$. In Figure 15 (left), shorter warmup lengths lead to higher possibilities of loss spikes This behavior is consistent with the sharp-to-flat dynamics. Early in training, the model resides in sharper regions of the landscape, where only sufficiently small LRs ensure stable updates. Therefore, warmup is needed to gradually increase the LR until the trajectory enters flatter regions that can tolerate larger LRs.

**What if zero warmup and small BS?** We also conduct experiments with no warmup and small batch size. In Figure 15 (right), the loss spikes become even more significant. This aligns with our

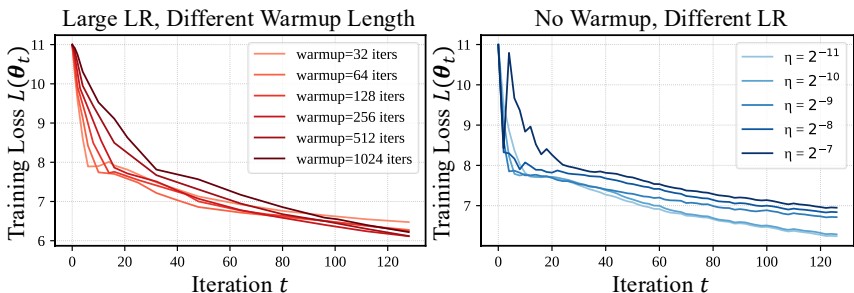

Figure 15: **(Left.) Shorter warmup, more loss spikes.** We train a series of `LLaMA-2` models with 170M parameters. We adopt a warmup-stable schedule, where the warmup length varies from $\{2^5, 2^6, 2^7, 2^8, 2^9, 2^{10}\}$ iterations and the peak LR is fixed $\eta = 2^{-7}$. **(Right). Zero warmup and small BS leads to larger loss spikes.** We train a series of `LLaMA-2` models with 170M parameters. We adopt a constant LR schedule, where no warmup and the peak LR is varied, $\eta \in \{2^{-11}, 2^{-10}, 2^{-9}, 2^{-8}, 2^{-7}\}$. We also use the 0.49M BS.

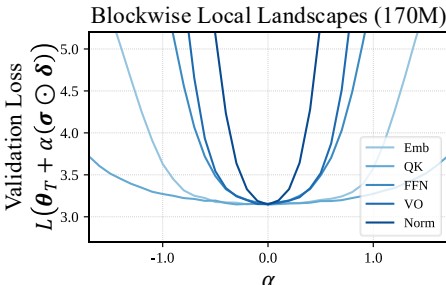

Figure 16: **Local loss landscape exhibits blockwise structure.** One-dimensional loss landscapes at the final iterate $\boldsymbol{\theta}_T$ along a masked random perturbation direction $\boldsymbol{\sigma} \odot \boldsymbol{\delta}$. Here, $\boldsymbol{\theta} \in \mathbb{R}^p$ is the random perturbation vector, and $\boldsymbol{\sigma}$ is a blockwise mask that zeros out perturbations outside the specified block type.

explanation: with no warmup, the LR jumps immediately to a large value while sharpness is still extremely high, causing a spike almost at initialization. More importantly, small BS does not replace warmup, and the instability still appears because of the sharpness.

## D.4 BLOCKWISE LANDSCAPE STRUCTURE

**Visualizing the Blockwise Local Loss Landscape.** Transformer architecture is composed of different block types, such as query–key (QK) and value–output (VO) projections, feedforward networks (FFN), normalization layers (Norm), and embedding layers (Embed). Prior studies (Zhang et al., 2024a; Wang et al., 2025) found that these block types exhibit heterogeneous levels of sharpness, suggesting that different block types contribute differently to the local loss landscape. To better understand this heterogeneity, we visualize the local loss landscape separately for each block type.

Similar to Figure 5, we perturb the final iterate $\boldsymbol{\theta}_T$ along a random direction $\boldsymbol{\delta}$, but restrict the perturbations to a selected block type using a blockwise mask $\boldsymbol{\sigma}$. In Figure 16 (right), local landscapes differ substantially across block types, and the curvature order we observe is QK < Embed < FFN < VO < Norm. This ordering is slightly different from the results reported by Wang et al. (2025), which found Embed < QK < FFN < VO < Norm. We suggest two possible reasons for this discrepancy. First, our analysis probes the most direction landscape, whereas Wang et al. (2025), following Wang et al. (2024), directly estimate sharpness from the fisher information matrix. Second, embeddings may not appear as the flattest block in terms of loss landscape, but as they are least activated during gradient propagation (many embedding entries receive no gradient), they are effectively flatter in training dynamics.

