# OpenReview forum: "How Does Local Landscape Geometry Evolve in Language Model Pre-Training?"
_ICLR.cc/2026/Conference — Submitted to ICLR 2026_

### Official Review · Reviewer_eiwb · 2025-10-23

**Soundness:** 3
**Presentation:** 3
**Contribution:** 3
**Rating:** 6
**Confidence:** 4

**Summary:**

This paper investigates how the local loss landscape geometry evolves during the pre-training of large language models. The authors identify two distinct phases in training dynamics.

Early phase: At the beginning of training, the loss landscape exhibits high sharpness. During this stage, the learning rate governs the training stability. Therefore, a **learning rate warmup** is essential to move from sharp to flatter regions safely.

Late phase: Once the model enters a more stable training regime, the batch size and the resulting gradient noise scale become the dominant factors shaping the local landscape. This implies the importance of a **dynamic batch-size schedule**.

**Strengths:**

**Systematic analysis**: The paper provides a systematic framework to analyze how the local loss landscape evolves during pre-training.

**Empirical insights**: It empirically identifies a two-phase transition in sharpness and links this dynamic to training stability and learning rate schedules.

**Practical implications**: The theoretical insights are translated into practical training strategies (learning rate warmup and dynamic batch-size schedule).

**Weaknesses:**

**Limited architecture and optimizer scope**: As noted in the paper’s limitations, the analysis is restricted to the LLaMA-2 architecture and the AdamW optimizer. It remains unclear whether the findings would generalize to other architectures or optimization algorithms.

**Lack of analysis on interaction effects**: The paper analyzes batch size ramping and learning rate decay in isolation. The interaction effects between BS ramping and LR decay therefore remain unclear.

**Theoretical simplifications**: Assumptions such as noise covariance being proportional to M, time-invariant M, equilibrium, and local quadratic approximation are introduced for theoretical convenience, but may not hold in actual training dynamics.

**Overstated novelty discussion**: The authors point out that prior works are “largely restricted to small-scale networks” and emphasize that their study provides the “first systematic study of how local landscape geometry evolves in large-scale language model pre-training.” To make this claim more meaningful, it would be helpful to more clearly articulate the qualitative differences between small-scale networks and large-scale language models. Moreover, discussing how the observed dynamics might change when scaling beyond the 93M and 170M parameter models used in this work would further strengthen the argument.

**Questions:**

Minor Errors

Line 129:  The Hadamard product are ->  The Hadamard product is

Line 132: near an local -> near a local

Line 134: at i-the example -> at i-th example

Line 146: gradient covariance at $\theta$ -> gradient covariance at $\theta^\star$

Line 157: Our experiments varies -> Our experiments vary

Line 67: landscapes gradually widens/flattens -> landscapes gradually widen/flatten

Line 266: in the most-case directions -> in the most directions

Line 279: This rational further suggests -> This rationale further suggests

Line 322: two key question remains -> two key questions remain

Line 172: Loss spikes and plateau -> Loss spikes and plateaus

Line 238:  only a sufficient small ->  only a sufficiently small

---

> ### Author Response · Authors · 2025-11-23
> **Response to Reviewer eiwb (1/2)**
>
> Thank you for your great efforts on the review of this paper and for recognizing the value of our contributions. We will try our best to address your questions.
>
> **If you still have further concerns, or if you are not satisfied by the current responses, please let us know, so that we can update the response ASAP.**
>
> **Q1: On the limited architecture and optimizer scope.** "As noted in the paper’s limitations, the analysis is restricted to the LLaMA-2 architecture and the AdamW optimizer. It remains unclear whether the findings would generalize to other architectures or optimization algorithms."
>
> **A1**: Thank you for this helpful suggestion. We have conducted **additional experiments** on alternative optimizers, such as `Lion` and `Adam-mini`, as well as on a different architecture `GPT-2`. Specifically, we evaluate the sharpness evolution in the early training iterations and validate our proposed BS scheduling method. We also verify the warmup-tuning recipe on the `GPT-2` architecture. These new results (see **Appendix D.2 and Fig. 9-12**) demonstrate that our main conclusions remain consistent across alternative architecture and optimizer choices.
>
> **Q2: On the interaction effect between LR decay and BS ramping.** "The paper analyzes batch size ramping and learning rate decay in isolation. The interaction effects between BS ramping and LR decay therefore remain unclear."
>
> **A2**: Thank you for this helpful suggestion. We have conducted **additional experiments** to explore the combined effect of LR decay and BS ramping. In revised **Fig. 8 (right)**, we compare four different schedules: at 10B tokens, a) LR drops to $4\times$ of its initial value; b) LR drops to $16\times$; c) BS ramps to $16\times$; d) LR drops to $4\times$ and BS ramps to $4\times$. Evidently, the schedules (b c d) , which preserve the ratio $\eta / B$, produce nearly identical loss curves; whereas schedule (a) results in a noticeably different trajectory.
>
> These results shows that any scheduler (including hybrid ones) that preserves $\eta/B$ exhibits nearly identical behavior. These results further validate the applicability of our theory, reinforcing our theoretical prediction that late-phase training dynamics are governed primarily by the noise scale $\tau \propto \eta / B$.
>
> **Q3: On the justification of theoretical simplifications.** "Assumptions such as noise covariance being proportional to M, time-invariant M, equilibrium, and local quadratic approximation are introduced for theoretical convenience, but may not hold in actual training dynamics."
>
> **A3**: Great point. We clarify each assumption as follows:
> - **Noise covariance assumption.** To obtain the SDE model, we assume $\frac{\eta}{B} \mathbf{M} \mathbf{\Sigma}(\boldsymbol\theta^\star) \mathbf{M}^\top = 2\tau \mathbf{M} + O(\eta)$. This assumption is not arbitrary: it follows directly from our discrete-time analysis in Lemma C.4. where we solve the variance of the linearized PSGD recursion in closed form. This assumption simply captures the regime in which the discrete stationary covariance remains finite as $\eta \to 0$. We have clarified this connection in our revision.
> - **Removal of the local quadratic assumption.** In response to your concern, we have removed the local quadratic assumption from the early-phase analysis. The revised Sec. 4 replaces it with a Lyapunov-based stability analysis, which does not require the loss to be locally quadratic and applies to general nonlinear dynamics. Further details are provided in **A1** for Reviewer fTYS and in the updated Sec. 4.
>
> These revisions make our theoretical arguments more general and reduce reliance on idealized assumptions.

---

> ### Author Response · Authors · 2025-11-23
> **Response to Reviewer eiwb (2/2)**
>
> **Q4: On the qualitative difference between small and large networks.** "To make this claim more meaningful, it would be helpful to more clearly articulate the qualitative differences between small-scale networks and large-scale language models."
>
> **A4**: Thank you for the suggestion. We'd like to clarify the qualitative differences between small-scale networks and large-scale language models:
>
> - **Task differences.** Small vision networks (e.g., ResNet-20 on CIFAR-10) are trained on relatively simple, low-diversity datasets where full interpolation is feasible. In contrast, LLM pre-training involves extremely large and heterogeneous text corpora, making the optimization regimes fundamentally different.
> - **Over- vs. under-parameterization**. Small networks are typically over-parameterized relative to their dataset and can drive the training loss to (near) zero. Pre-trained LLMs, however, are effectively under-parameterized relative to the scale of their data. Even after extensive training, the training loss remains large, and validation loss closely matches training loss, indicating minimal overfitting.
> - **Generalization vs. convergence**. Because small networks can easily fit the training set, the key question becomes generalization, motivating optimization algorithms seeking for flat minima. In LLM pre-training, however, the challenge is convergence, i.e., reducing the loss as much as possible under a fixed data budget. Therefore, we choose to deepen the basin using BS scheduling, rather than widen it as in the classical flat-minima perspective.
>
> **Q5: On the limited model size in experiments.** "Moreover, discussing how the observed dynamics might change when scaling beyond the 93M and 170M parameter models used in this work would further strengthen the argument."
>
> **A5**: Thank you for this helpful suggestion. We have conducted **additional experiments** on larger models (270M and 530M parameters) to further strengthen our claims. Due to the high computational cost of Hessian-based analysis, we currently validate only the BS scheduling method on these larger scales; we are still working on extending the sharpness-evolution analysis.
>
> The new results (**see Appendix D.2 and Fig. 13**) show that our BS scheduling strategy becomes **even more effective as model size increases**, further supporting the robustness of our conclusions across scales.
>
> **Q6: Typos and other minor issues.**
>
> **A6**: Thank you for your careful review. Your feedback is invaluable in helping us improve our paper. We have carefully revised the paper to address the typos you highlighted.

---

> ### Author Response · Authors · 2025-11-26
> **Looking forward your feedback**
>
> Dear Reviewer eiwb,
>
> We hope that our responses adequately address your concerns. As the deadline of this discussion phase is approaching, we warmly welcome further discussion regarding any additional concerns that you may have.
>
> Thank you for the time and appreciation that you have dedicated to our work.
>
> Best regards,
>
> Authors of submission 9645

---

### Official Review · Reviewer_G3Yh · 2025-10-29

**Soundness:** 2
**Presentation:** 2
**Contribution:** 1
**Rating:** 2
**Confidence:** 3

**Summary:**

The authors analyze the "local landscape" of the loss as training progresses. The paper is organized around two phases: "early" (when loss is sharp) and "late" when loss is flat. Within these two phases, they ask and answer two questions related to training dynamics. Based on these theoretical results, they offer heuristics/principles which they instantiate and show experimentally.

The questions/answers are:

* Why do short warmups produce instability? (Because the LR rises too quickly when the model is still too sharp)
* Why do spike and plateaus only happen at the beginning of training? [Reviewer's note: they do not only occur then?] Because things are smoother later on.
* "Why is there a trade-off between widening and deepening the basin?" because temperature affects basin selection and different temperatures prefer different basins.
* How does BS impact this trade-off? BS impacts temperature.

**Strengths:**

At a high level, the paper reads really nicely. It's easy to follow: the math, the figures, the simulation results, and the real results are all presented in a coherent, digestible way. It would, in many ways, make a nice tutorial/review about some of the concepts in this space, linking theory to practice.

I especially like the visualizations of sharpness along 1D slices. This is a nice visual tool, though I feel like the authors don't actually make that much use of it.

I think the experiments mostly support the claims made in the paper (mostly, though I have a quibble).

**Weaknesses:**

While it is nicely written, the authors claim much too much novelty. They apply existing tools to familiar settings but without much new insight. They don't seem to arrive at any substantially different conclusions. The math is sometimes a bit different (though usually not), but when it is, it provides seemingly no insight over what's already known.

Given that there is in fact quite a lot of prior art for many of these findings (which, admittedly, the authors do frequently cite), it would be good if they empirically compared their prescriptions to those other sources.

## Most results are known, both theoretically and empirically

To be blunt, I feel like the majority of the insights of this paper are fairly well known, both from an empirical and a theoretical perspective.
For example, I think theorem 5.1 is a repackaging of known results (e.g. Jastrzębski et al. (2018) eqn 12 is I think the two-basin version of this).

To drive the point home, I asked ChatGPT: "does the batch size impact the type of basin found when using Adam?" and it produced a detailed, similar explanation with cites to existing theoretical and empirical work." (To be clear, I am not insinuating the authors had ChatGPT write this paper, just that the results are standard.)

I could dig up additional academic sources, but I'm fairly confident that I could pose ChatGPT most of the questions addressed in this paper and reach largely similar results, getting already published papers, both empirical and theoretical.

As another example, it's well known that BS and LR trade off with one another, including using theory from SDEs! One example: https://www.cs.princeton.edu/~smalladi/blog/2024/01/22/SDEs-ScalingRules/

The batch size warmup is also known, with known theoretical grounding relating to CBS/gradient noise scale: https://arxiv.org/pdf/2505.23971

And even the "LR warmup needs to be longer for higher max LRs" is known, including a similar theoretical justification. The authors dismiss Kalra and Barkeshli  as being focused on resnets, but they also do studies on GPT-2 style transformers with much longer warmups than the authors claim in their description of the work?

## The principles/recipes are no more actionable than what is known/done

The "recipe" for LR warmup length literally just says... "the larger the peak LR, the longer the warmup should be". How much longer? proportional to LR change? proportional to LR^2?

To me, a recipe would suggest a particular heuristic or criterion for, say, warming up the batch size. The authors say "Ramp the BS once loss reduction becomes marginal." but then seemingly the authors just use a fixed step/token count for when to ramp? How is this an improvement over what we know? (CF Merill et al 2025 https://arxiv.org/pdf/2505.23971 which actually do provide a criterion for when to scale that can be tracked during training)

**Questions:**

To be a bit less polemical, regarding the batch size experiments: the BS vs LR doesn't really match up super closely: increasing BS and decreasing LR both make the loss go down, great. The Princeton link I pasted above would say that Adam LR should scale with sqrt(BS) rather than linearly. If you ran your experiment using sqrt(BS) scaling, would it match more closely? I guess yes.

Merrill et al explore batch size warmup from a CBS perspective, deriving a specific, trajectory-driven way of determining when to warm up batch size. How does your approach compare?

---

> ### Author Response · Authors · 2025-11-23
> **Response to Reviewer G3Yh (1/3)**
>
> Thank you for your great efforts on the review of this paper. We will try our best to address your questions.
>
> **If you still have further concerns, or if you are not satisfied by the current responses, please let us know, so that we can update the response ASAP.**
>
> **Q1: On the novelty of our contribution in the late phase part.** "I feel like the majority of the insights of this paper are fairly well known, both from an empirical and a theoretical perspective. .., I think theorem 5.1 is a repackaging of known results (e.g. Jastrzębski et al. (2018) eqn 12 is I think the two-basin version of this).” and “The batch size warmup is also known, with known theoretical grounding relating to CBS/gradient noise scale."
>
> **A1**: Thank you for the thoughtful feedback. We now explicitly **acknowledge** Jastrzębski et al. (2018) in the revised Sec. 5. Their SDE-based analysis indeed provides an early form of our Thm. 5.1. However, we'd like to highlight the novelty and contributions of this paper.
>
> 1. **Theory is not the main contribution.** Our theory is meant to *explain* the empirical observations and to motivate a practical BS scheduler. We choose SDE modeling because it is a simple and powerful tool, widely used in prior works, as noted by the reviewer. Our goal is not to introduce a fundamentally new theoretical framework but to use standard tools to clarify the mechanisms underlying our findings; developing new theory is a separate interest outside the scope of this paper.
> 2. **Different insights from Jastrzębski et al. (2018).** While both works use SDE modeling, the focus and insights differ. Jastrzębski et al. (2018) show that the ratio $\eta / B$ influences generalization, training dynamics, and the width of minima found by SGD. In contrast, our work uses the SDE modeling to motivate a practical, dynamic BS scheduling rule. For example, we argue that training should start with a small BS. Because, according to Thm. 5.1, the loss term dominates the depth–flatness trade-off in the early phase, when large BS provides little benefit while significantly increasing data consumption.
> 3. **Our BS scheduling is not explained by CBS or BS warmup.** A key misunderstanding is that our scheduling is equivalent to CBS-based BS warmup. CBS focuses on balancing training time and data efficiency, and BS warmup methods (e.g., Merrill et al., 2025) increase BS once CBS exceeds the current BS. In contrast, our scheduling is **fundamentally different**: we choose to **ramp BS only very late in training**, when the loss reduction becomes marginal. This scheduling is based on two empirical discoveries (previously not included in the paper).
>     -  All BS ramping schedules eventually collapse to the same loss trajectory (**Fig. 6 left**). Early ramping offers no efficiency benefit, achieving the same loss but with higher data consumption.
>     -  Under a fixed token budget, late BS ramping consistently performs best (**Fig. 6 right**).
>
>     These findings reveal a behavior absent from CBS-related literature: **Since all schedules converge to the same trajectory, delaying BS ramping allows maximal progress (lower loss) under a data-limited budget**. This insight motivate our Design Principle II, an insight not predicted by CBS theory.
>
> **In summary**, we fully acknowledge the connection to earlier SDE theory, but our empirical observations and proposed BS scheduling are not captured by prior work on CBS or SDE theory.
>
> **Q2: On the interplay between BS and LR.** "it's well known that BS and LR trade off with one another, including using theory from SDEs." and "Adam LR should scale with sqrt(BS) rather than linearly. If you ran your experiment using sqrt(BS) scaling, would it match more closely? I guess yes."
>
> **A2**: Thank you for this question.
> - **Not a BS-LR trade-off, but a depth-width trade-off.** Our SDE analysis shows that it is the ratio $\eta / B$ controlling the trade-off between deepening and widening the basin. Similar insights are provided in Jastrzębski et al. (2018), but our formulation (Thm. 5.1) provides an explicit free-energy expression that directly reflects the tradeoff and further motivate our BS scheduling.
> - **Empirical test of $\sqrt{B}$ scaling.** Following your suggestion, we have conducted **additional experiments** comparing:
>     1. At $T_{ramp}$ tokens, ramping the BS by $x$.
>     2. At $T_{decay}$ tokens, decaying the LR by $\sqrt{x}$.
>
>     We choose $x=16$ and $T_{ramp}=T_{decay}=10$B tokens. In **Fig. 8 (right)**, we observe:
>     1. Decaying LR by $\sqrt{16}=4\times$ produces a **significantly different trajectory** from $16\times$.
>     2. Ramping BS by $16\times$, decaying LR by $16\times$, and even simultaneously ramping BS by $4\times$ and decaying LR by $4\times$, all produces **nearly identical loss curves**.
>
>     Therefore, rather than following $\sqrt{B}$ scaling, these results confirms our theory that training dynamics are governed primarily by the noise scale $\tau \in \eta / B$.

---

> ### Author Response · Authors · 2025-11-23
> **Response to Reviewer G3Yh (2/3)**
>
> **Q3: On the novelty of our contribution in the early phase part.** "…'LR warmup needs to be longer for higher max LRs' is known, including a similar theoretical justification. The authors dismiss Kalra and Barkeshli as being focused on resnets, but they also do studies on GPT-2 style transformers with much longer warmups than the authors claim in their description of the work?"
>
> **A3**: Thank you for this question.
> - **Is "LR warmup needs to be longer for higher max LRs" already known?** This is not an estabilished result in prior works. While early works, including Kalra & Barkeshli (2024), note that warmup enables the use of larger target LRs, no prior work shows that the optimal warmup length scales *proportionally* with the peak LR within a reasonable range.
>     - Kalra & Barkeshli (2024) showed that warmup allows larger $\eta_{trgt}$ by steering the training into better-conditioned regions, but for a fixed $\eta_{trgt}$ they repeatedly emphasize that extending warmup provides only marginal benefits.
>     - Moreover, existing LLMs (e.g., GPT, OPT, etc) use fixed warmup schedules independent of peak LR.
> - **How does our work differ from Kalra & Barkeshli (2024)?** Although both papers study the sharpness, the focus and conclusion differ:
>     - Kalra & Barkeshli (2024) study small-scale models (FCNs/ResNets) under full-batch Adam. They do not measure sharpness evolution for GPT-style transformers under standard AdamW pre-training setup.
>     -  Their interpretation of early training is based on the catapult/self-stabilization regimes. They does not discover the sharp-to-flat dynamics in the early phase of LLM pre-training.
>
>     Our work provides the *first direct visualization* of Sharpness evolution during realistic LLM pre-training and identifies a sharp-to-flat transition that directly implie the proportional warmup rule.
>
> Overall, we believe our results are new and not contained in prior works, especially from the perspective of loss landscape geometry evolution and LLM pre-training.
>
> **Q4: On the tuning recipe for LR warmup length.** "The
> 'recipe' for LR warmup length literally just says... 'the larger the peak LR, the longer the warmup should be'. How much longer? proportional to LR change? proportional to LR^2?"
>
> **A4**: Yes, **proportional to LR change**. In our original Fig. 5, the curves suggested a proportional relationship between peak LR and optimal warmup length. However, as other reviewer noted, this proportionality does not hold universally. To avoid misinterpretations, we have replaced the Fig. 5 with a new figure (now **Fig. 4**).
>
> The revised figure explicitly shows that larger peak LRs consistently require longer warmup, but the proportional scaling only holds within a LR range of $2^{−8}$ to $2^{−11}$. Outside this range, increasing the warmup further does not improve (nor meaningfully worsen) the final loss; warmup simply becomes less sensitive.
>
> We further verify this proportionality on **GPT-2 architectures**. In **Appendix D.2. and Fig. 10**, these new results remains consistent with our main findings.

---

> ### Author Response · Authors · 2025-11-23
> **Response to Reviewer G3Yh (3/3)**
>
> **Q5: On the novelty of BS scheduling and comparison with Merill et al 2025.** "To me, a recipe would suggest a particular heuristic or criterion for, say, warming up the batch size. The authors say "Ramp the BS once loss reduction becomes marginal." but then seemingly the authors just use a fixed step/token count for when to ramp? How is this an improvement over what we know? (CF Merill et al 2025 https://arxiv.org/pdf/2505.23971 which actually do provide a criterion for when to scale that can be tracked during training)"
>
> **A5**: Thank you for raising this important point.
> - **How is this an improvement over what we know? Ramping BS late in training.** Current BS scheduling methods mostly rely on the CBS theory, including Merill et al. (2025). CBS is estimated via gradient noise scale. Because gradient noise increases over training, CBS also increases and these method claims to ramp the BS accordingly. In contrast, our method differs:
>     - **Different goals.** CBS-related BS scheduling is mainly designed fro training-time efficiency while preserving data efficiency. Instead, our focus is the data limited regime. We aim to **achieve the lowest possible loss under a fixed token budget**. In this setting, CBS theory does not give direct guidance on *when* BS should increase to maximize loss reduction per token.
>     - **Different method.** Instead of ramping BS according to CBS, our second design principle suggest to **ramp the BS only very late in training**, until the loss reduction becomes marginal. Our method is based on two interesting discoveries:
>         - All BS ramping schedules converge to the same loss trajectory (**Fig. 6 left**). Early ramping provides zero improvement in loss and only increases token consumption.
>         - Under a fixed token budget, late ramping consistently performs best (**Fig. 6 right**).
>
>         These observations cannot be explained via CBS theory. In certain cases, **our finding even suggest the opposite**: in the data-limited regime, early ramping is harmful, even when CBS would permit it.
>
> - **Our "criterion" is conceptual, not tied to fixed token positions.** Our design principle, ramp BS once loss reduction becomes marginal, is not a fixed step count; it is based on identifying the transition between the loss-dominated and noise-dominated phases. In our experiments, for clarity, we place the ramp at fixed positions to isolate the effect of timing. But the principle does not depend on a predetermined count. In practice, one could track marginal loss reduction to decide when this transition occurs. Our paper focuses on discovering the design principle, not designing a new online estimator.

---

> ### Author Response · Authors · 2025-11-26
> **Looking foward your feedback**
>
> Dear Reviewer G3Yh,
>
> We hope that our responses adequately address your concerns. As the deadline of this discussion phase is approaching, we warmly welcome further discussion regarding any additional concerns that you may have, and we sincerely hope you can reconsider the rating accordingly.
>
> Thank you for the time and appreciation that you have dedicated to our work.
>
> Best regards,
>
> Authors of submission 9645

---

### Official Review · Reviewer_6uFu · 2025-10-31

**Soundness:** 3
**Presentation:** 3
**Contribution:** 3
**Rating:** 4
**Confidence:** 3

**Summary:**

The paper admits a two-phase analysis on language model pretraining, to address two crucial decisions required: the learning rate warmup phenomenon and batch-size scheduling.
The authors note the common hyperparameter settings used in the literature, wherein a large number of warmup steps are used for the maximal stable learning rate (LR) applied, followed by a stable or annealing LR.
Also, the use of arbitrary batch size schedules is reported in the literature.
The locally quadratic loss landscape analysis in the paper looks at two distinct training stages relying on SGD: (i) *early phase*: where the loss landscape is sharper and the imperative is `to flatten`, and thereby LR warmup, to begin optimization from a less sharp region, or attractor basins; (ii) *late phase*: where the loss landscape tends to a flatter region owing to SGD convergence, and the imperative is `to deepen` the loss basin reached for an improved loss, which is done by either LR decay or a batch size ramp up.

**Strengths:**

* Strong motivation and setup, to address two important empirical choices in language model pretraining, which are often the first hyperparameters adjusted for any new task, hardware.

* Fairly clear math and assumptions for the analytical explanation of the pre-training dynamics using the quadratic approximation of the loss landscape around the theoretical minima; empirical analysis supporting claims.

* Clear conditions given on the practical rule-of-thumb in setting the LR warmup and Batch size ramping, for a warmup-stable learning rate schedule.

**Weaknesses:**

* The terminology and goal of the analysis can be made cleaner; that is, the terms: sharpness, flat minima, wide basin, deep basin could be clarified independently early on before the lemmas and theorems.
  * It is hard to understand if the direction of analysis emerges as a result of wanting to go from sharp to flat early on (and wide to deep in the later phase), or vice versa. Especially since the locally quadratic assumption does not necessarily hold at initialization.

* Given LR decays are an important discussion in managing LR schedules and thus scaling and the laws fit on this data, the interaction of both LR decay and increasing batch size feels underexplored.

* Given the relatively limited empirical experimental scope, a broader *Limitations* section is warranted: especially for the unknowns such as the role of LR annealing choice; how the timing and the number of batch ramp ups matter.

**Questions:**

Below is an enumeration of various questions and suggestions.

Please note that the rating will go up contingent on the points below, with more weightage on the following points: 1, 2, 4, 5, 9.

1\. L76-81: Could the authors please explain (or elaborate here) why the deepening of the loss basin at *late-stage* training is crucial and different from the flat-minima conversation around SGD convergence?

2\. L106-114: Could the authors include [1] here and also contrast their early-late training phase interpretation with the loss landscape perspective given here?

3\. Equation 3: What is the role or effect of the $B$ in the denominator given the $1/n$ already included in $\Sigma ({\theta}^{*})$ (L145-146)?

4\. Figure 2: It appears that only the larger LRs (in the grid shown here) lead to spikes. Could the authors intuit why and how the finding here regarding the edge-of-stable LRs explains this phenomenon?

* How does zero-warmup (not in Figure 2) actually affect the findings here, and also the actual effect on *moving away from sharp loss landscapes*? Given we expect noisy updates in the beginning few update steps in most cases, does no warmup and a small enough batch size have the same effect?

5\. L178: Could the authors make an overall comment on how the absence of LR decay influences the assumptions and findings, and thereby the practical recommendations? Given that recent literature suggests LR decay is crucial for a model before modeling its loss obtained for scaling law derivations, downstream performances, finetuning, etc.

6\. Figure 3: Could the authors please provide some guidance on how to read Figure 3? What exactly is being plotted with the orange line, and what are $u_1$ and $u_w$?

7\.1. Figure 4 (top): Which layer matrix are the eigenvalues being reported?

7\.2. Figure 4 (bottom): Does this perform a perturbation on all the weights?

8\. Lemma 4.2: Could the authors explain (or intuit) the definition of $z$ and therefore $z_l$?

9\. L270-291 and Figure 3: Does the figure suggest that anything *more* than the optimal LR warmup length leads to a worse loss? What, thus, is a *reasonable range* (L290) in practice?

10\. Figure 10: Could the settings here be marked differently (markers/linestyles) and not just with transparency?

11\. Personally, took me a long time to understand why we want to move away from a wide basin, until I realized we are talking about a region around the minima (a basin) and not the flat minima we converge to. Therefore, the finding of a small batch required for a wider basin felt counterintuitive and required re-reading of Section 5. The motivation, analysis, and conclusion can be presented much more cleanly, building up to a general batch size schedule. Additional comments on this or identifying future directions w.r.t. role of LR schedules could be more explicit.

12\. L266, L469:  What does `most-case direction` mean?

---

References:

[1] Straight to Zero: Why Linearly Decaying the Learning Rate to Zero Works Best for LLMs, Bergsma et al., 2025, arXiv:2502.15938 [cs.LG].

---

> ### Author Response · Authors · 2025-11-23
> **Response to Reviewer 6uFu (1/4)**
>
> Thank you for your great efforts on the review of this paper. We will try our best to address your questions.
>
> **If you still have further concerns, or if you are not satisfied by the current responses, please let us know, so that we can update the response ASAP.**
>
> **Q1: Suggestions for more clarification of terminologies.** "The terminology and goal of the analysis can be made cleaner; that is, the terms: sharpness, flat minima, wide basin, deep basin could be clarified independently early on before the lemmas and theorems."
>
> **A1**: Thank you, we agree that clearer terminology improves readability. We have added explicit definitions in **Appendix A**:
>
>
> | Terminology | General Meaning | Usage in This Paper |
> | -- | -- | -- |
> | Sharpness| A measure of curvature in the loss landscape, often characterized via the Hessian. Different works may define it differently. | We define sharpness as the curvature along the sharpest direction of the loss landscape. Mathematically, it is presented as the largest eigenvalue of the Hessian $\lambda_{max}(\mathbf{H}(\boldsymbol\theta_t))$ or of the preconditioned curvature matrix $\lambda_{max}(\mathbf{S}(\boldsymbol\theta_t))$. |
> | Flat/sharp minimum | A minimum is a point where the gradient vanishes $\nabla L(\boldsymbol{\theta})=0$ and the loss does not decrease in small neighborhood. A sharp minimum has large curvature; a flat minimum has small curvature. | We use these terms sparingly and follow the standard definitions from the sharpness/flat-minima literature. |
> | Wide/deep basin | A loss basin is a region of the landscape surrounding a minimum. A wide basin rises loss slowly in most directions, whereas a deep basin has a significantly lower minimum value compared to its surroundings. | We use these terms to establish the depth–flatness trade-off: large noise scales tend to find wide basins, while small noise scales tend to find deeper regions with lower loss. |
>
>
> **Q2: On the validity of the local quadratic assumption.** "It is hard to understand if the direction of analysis emerges as a result of wanting to go from sharp to flat early on (and wide to deep in the later phase), or vice versa. Especially since the locally quadratic assumption does not necessarily hold at initialization."
>
> **A2**: Thank you for your question. Our analysis follows the actual steps of our investigation rather than a predetermined narrative:
> 1. **Motivating observations.** We first observed that loss spikes consistently appear when warmup is shortened or the peak LR is increased. We then ruled out alternative explanations such as BS choices and optimizer instability (see **Appendix D.4 and Fig. 14-15**).
> 2. **Theory-inspired explanations.** These observations naturally led us to theoretical stability analysis, which is tied to the curvature of the loss landscape. We hypothesized that the instability is driven by large sharpness early in training, and we verified this empirically: sharpness is highest at initialization and decreases as training progresses. This sharp-to-flat evolution also explains why longer warmup avoids spikes even with large LRs.
>
> In addition, we have **revised our theory in Sec. 4**. Rather than relying on a local quadratic assumption, we now adopt a more general Lyapunov-based stability analysis that does not require quadratic structure. See **A1** for Reviewer fTYS and the updated Sec. 4 for further details.
>
> **Q3: On the interplay between LR decay and BS ramping.** "Given LR decays are an important discussion in managing LR schedules and thus scaling and the laws fit on this data, the interaction of both LR decay and increasing batch size feels unexplored."
>
> **A3**: Thank you. Following your suggestion, we have conducted **additional experiments** to explore the combined effect of LR decay and BS ramping. In **revised Fig. 8 (right)**, we compare four different schedules: at 10B tokens, a) LR drops to $4\times$ of its initial value; b) LR drops to $16\times$; c) BS ramps to $16\times$; d) LR drops to $4\times$ and BS ramps to $4\times$. Evidently, the schedules (b c d) , all of which preserve the ratio $\eta / B$, produce nearly identical loss curves; whereas schedule (a) results in a noticeably different trajectory.
>
> These results shows that any scheduler (including hybrid ones) that preserves $\eta/B$ exhibits nearly identical behavior, reinforcing our theoretical prediction that late-phase training dynamics are governed primarily by the noise scale $\tau \propto \eta / B$.

---

> ### Author Response · Authors · 2025-11-23
> **Response to Reviewer 6uFu (2/4)**
>
> **Q4: Suggestions for a broader limitation section.** “Given the relatively limited empirical experimental scope, a broader Limitations section is warranted: especially for the unknowns such as the role of LR annealing choice; how the timing and the number of batch ramp ups matter.”
>
> **A4**: Great suggestion. We have added **a broader Limitations** section in the **Appendix B** to address these concerns, including discussions of hyperparameter choices and architectural impacts. We also address several of the limitations directly in this rebuttal:
> - **Limited experimental scope.** We haved conducted additional experiments on larger models, alternative optimizers, and different architectures. In **Appendix D.2**, we examine the sharp-to-flat dynamics and validate our BS scheduling method under these new setups. These results show that our main conclusions remain consistent across larger scales, optimizers and architectures.
> - **LR annealing choices.** We clarify that the warmup-stable schedule used in the main paper is intended to isolate the effect of BS ramping from LR decay. As detailed in **A3**, we have evaluated the combined effect of LR decay and BS ramping, further validating our theory. We acknowledge that additional LR schedules could be explored; we discuss this in the expanded Limitations section and view it as an important direction for future work.
> - **Timing of BS rampings**. We conducted **new experiments** to examine the effect of ramping timing. In the **Fig. 6 (left)**, we find that all loss curves eventually collapse onto the same trajectory regardless of when BS ramping occurs, suggesting that delaying the ramp allows maximal progress along this trajectory under a data-limited budget. More directly, in **Fig. 6 (right)**, we observe that under fixed token budget, BS ramping is most effective when applied late in training. These new findings further support our design principle II, which recommends increasing the BS only once loss reduction becomes marginal.
> - **Number of BS ramping**. In **Fig. 7**, it is evident that While each BS ramping step still offers a measurable improvement in val. loss, additional ramp-ups lead to diminishing returns. This suggests that, beyond a small number of well-timed BS rampings, further increases in BS provide limited benefits and may not be cost-effective under a data-limited budget.
>
>
> **Q5: On the difference between deepening the basin and flat-minima hypothesis.** "L76-81: Could the authors please explain (or elaborate here) why the deepening of the loss basin at *late-stage* training is crucial and different from the flat-minima conversation around SGD convergence?"
>
> **A5**: Great question. The key difference comes from the training regime. In standard vision settings (e.g., ResNet-20 on CIFAR-10), the models are **over-parameterized** and can nearly interpolate the data. Training loss goes to zero, so the main concern is generalization, making the flat-minima hypothesis relevant.
>
> In contrast, LLM pre-training is **under-parameterized** relative to the scale of data. After one pass, the training loss remains high, and the val. loss is almost identical to the training loss, indicating minimal overfitting. Thus, before generalization matters, the priority is to drive the loss lower, which requires reaching deeper basins.
>
> Our analysis focuses on how reduced noise (e.g., via larger batch size) helps the optimizer move into deeper minima, conceptually different from the flat-minima perspective in fully interpolating regimes.
>
> **Q6: Suggestions for comparison with [1].** "L106-114: Could the authors include [1] here and also contrast their early-late training phase interpretation with the loss landscape perspective given here?"
>
> **A6**: Thank you for the suggestion. We have added [1] and clarified the conceptual contrast.
>
> The key difference is that our early phase is driven by a sharp-to-flat transition in the local loss landscape. Sharpness is extremely high at the beginning, so the LR must stay small until this sharpness decays. This is why LR warmup is necessary. In contrast, [1] attributes the early phase to bias reduction rather than curvature.
>
> In the late phase, both works consider the role of gradient noise, but our interpretation differs. [1] argues that LR decay is needed to reduce variance and stabilize updates. However, our work shows that noise controls a depth–flatness trade-off: smaller noise drives toward deeper basins, while larger noise favors wider basins. Moreover, we show that reducing the noise scale late in training remains effective, leading to our BS scheduling that, in the data-limited regime, the BS should be ramped very late in training.

---

> ### Author Response · Authors · 2025-11-23
> **Response to Reviewer 6uFu (3/4)**
>
> **Q7**: "Figure 2: It appears that only the **larger LRs** (in the grid shown here) lead to spikes. Could the authors intuit why and how the finding here regarding the edge-of-stable LRs explains this phenomenon? How does zero-warmup (not in Figure 2) actually affect the findings here, and also the actual effect on *moving away from sharp loss landscapes*? Given we expect noisy updates in the beginning few update steps in most cases, does no warmup and a small enough batch size have the same effect?"
>
> **A7**: Thank you.
>
> - **Large LR alone do not cause loss spikes.** First, we clarify that large LRs alone do not cause loss spikes. In **Fig. 15 (left)**, we show loss curves under various warmup lengths with a fixed large peak LR. The results clearly show that only short warmup leads to spikes; with sufficient warmup, even large LRs remain stable.
>
> - **Why sharpness evolution explains the phenomenon.** It is well known that LR interacts with sharpness to determine stability. In a simple quadratic model where the Hessian (and thus sharpness) is fixed, there are only two cases:
>     - LR too large relative to sharpness → divergence
>     - LR sufficiently small → convergence
>
>     However, in our experiments, instability occurs only when the LR rises too quickly early in training, not simply when LR is large. This suggests that sharpness must be evolving. Indeed, we observe that sharpness is initially high and then gradually decreases. This explains everything:
>     - Initially, sharpness is high → LR must remain small → short warmup causes spikes.
>     - Later, sharpness becomes low → the same large LR becomes stable → no further spikes.
>
> - **Regarding zero warmup and small BS.** We have conducted **additional experiments** with no warmup and small batch size. In **Fig. 15 (right)**, the loss spikes become even more significant. This aligns with our explanation: with no warmup, the LR jumps immediately to a large value while sharpness is still extremely high, causing a spike almost at initialization. More importantly, small BS does not replace warmup, and the instability still appears because of the sharpness.
>
> Together, these results confirm that the early loss spikes are driven by rapid LR increases when the landscape is still sharp, not by LR magnitude alone or optimizer/BS choices.
>
> **Q8**: "L270-291 and Figure 3: Does the figure suggest that anything *more* than the optimal LR warmup length leads to a worse loss? What, thus, is a *reasonable range* (L290) in practice?"
>
> **A8**: In the original Fig. 5, the curves suggested a proportional relationship between peak LR and optimal warmup length. However, as you noted, this proportionality does not hold universally. To avoid misinterpretations, we have replaced the Fig. 5 with a new figure (now **Fig. 4**).
>
> The revised figure explicitly shows that larger peak LRs consistently require longer warmup, but the proportional scaling only holds within a LR range of $2^{-8}$ to $2^{-11}$. Outside this range, increasing the warmup further does not improve (nor meaningfully worsen) the final loss; warmup simply becomes less sensitive. Thus, the "reasonable range" refers to this LR interval where warmup length has a predictable, monotonic effect on performance.
>
> **Q9:** "Equation 3: What is the role or effect of the $B$ in the denominator given the $1/n$ already included in $\Sigma(\theta^*)$ (L145-146)?"
> **A9**: The $1/n$ in $\Sigma(\boldsymbol\theta)$ and $1/B$ play different roles:
> - $\Sigma(\boldsymbol\theta)$ is the population gradient covariance, defined over the entire dataset. The factor $1/n$ normalizes the covariance across all $n$ samples.
> - In contrast, the factor $1/B$ arises because the usage of mini-batch. When we average the gradients of $B$ samples, the variance scales as $Var(\frac{1}{B}\sum_{i\in B} \nabla L_i(\boldsymbol{\theta}))=\frac{1}{B}\Sigma (\boldsymbol{\theta})$.
>
> Therefore, the effective noise of SGD is proportional to $1/B$, and larger BS leads to smaller gradient noise.
>
> **Q10**: "L178: Could the authors make an overall comment on how the absence of LR decay influences the assumptions and findings, and thereby the practical recommendations?"
>
> **A10**: Thank you. We have added new experiments to study the interplay between LR decay and BS ramping. As detailed in **A3**, LR decay reduces the noise scale in the same way as BS ramping, and any schedule—including hybrid schedules combining both—that preserves $\eta / B$ produces nearly same loss curves. These new results further validate our theory.
>
> In practice, we recommend using LR decay together with BS scheduling, as both reducing the noise scale in the late phase of training.
>
> **Q11**: "Figure 3: Could the authors please provide some guidance on how to read Figure 3? What exactly is being plotted with the orange line, and what are $u_1$ and $u_2$ ?"
>
> **A11**: Thank you. Figure 3 has been removed in our revision due to updates to the theory in Sec. 4.

---

> ### Author Response · Authors · 2025-11-23
> **Response to Reviewer 6uFu (4/4)**
>
> **Q12**: "Figure 4 (top): Which layer matrix are the eigenvalues being reported?"
>
> **A12**: The eigenvalues in Fig. 4 (top) are computed from the **full-parameter Hessian**. This is computationally expensive, and due to resource limitations, we are only able to compute sharpness for models up to 170M parameters.
>
> **Q13**: "Figure 4 (bottom): Does this perform a perturbation on all the weights?"
>
> **A13**: Yes. The perturbations in Fig. 1, 4 and 5 are applied to all model weights.
>
> **Q14**: "Lemma 4.2: Could the authors explain (or intuit) the definition of $z$ and therefore $z_l$?"
>
> **A14**: Thank you. In the original paper, $\boldsymbol{z}$ denoted the projection of the vector $\boldsymbol{e}$ onto the eigenbasis of the preconditioned curvature matrix, and $z_{\ell}$ referred to its $\ell$-th coordinate. This change of variables diagonalized the preconditioned dynamics (Eq. 4), allowing each coordinate to evolve independently. In the revised version, Lemma 4.2 no longer relies on this transformation, so $\boldsymbol{z}$ is no longer needed.
>
> **Q15**: "Figure 10: Could the settings here be marked differently (markers/linestyles) and not just with transparency?"
>
> **A15**: Thank you for the suggestion, but could you clarify what you mean by "marked differently with markers/linestyles and not just with transparency"? We would be happy to adjust the figure accordingly once we better understand the concern.
>
> **Q16**: **On the presentation of this work.** "The motivation, analysis, and conclusion can be presented much more cleanly, building up to a general batch size schedule. Additional comments on this or identifying future directions w.r.t. role of LR schedules could be more explicit."
>
> **A16**: Thank you for this suggestion. We have substantially revised Sec. 5 to present the motivation, theoretical analysis, and the BS scheduling in a clearer progression:
> 1. **Motivating observations.** We first observe that large BS tends to find deeper basins; whereas small BS tends to reach wider basins.
> 2. **Connection to theory.** Via SDE analysis, we then attribute this behavior to the depth-flatness tradeoff governed by the noise scale $\tau \propto \eta / B$.
> 3. **BS Scheduling** Because LLM pre-training operates in a data-limited regime and aims to minimize training loss, an effective BS scheduler is needed.
>     a. **Design Principle I.** By the theory, early in training, the loss term dominates the trade-off; thus, we begin with a small batch size.
>     b. **Design Principle II.** After analyzing the timing of BS ramping, we choose to ramp the BS late in training, when the loss reduction becomes marginal.
>
> We also added comments in the **new limitations section (Appendix B)**, discussing the role of LR schedules and outlining future directions.
>
> **Q17**: "L266, L469: What does most-case direction mean?"
>
> **A17**: We follow the terminology "most-case direction" from Chen et al. (2025). Technically, it refers to applying a random perturbation to the model weights. Such directions are considered "most-case" because, in high dimensions, a random vector is likely to represent a typical direction rather than a special or structured one (e.g., an eigenvector).
>
> Now we have changed "most-case direction" to "most direction" to avoid misunderstandings.

---

> ### Author Response · Authors · 2025-11-26
> **Looking forward your feedback**
>
> Dear Reviewer 6uFu,
>
> We hope that our responses adequately address your concerns. As the deadline of this discussion phase is approaching, we warmly welcome further discussion regarding any additional concerns that you may have, and we sincerely hope you can reconsider the rating accordingly.
>
> Thank you for the time and appreciation that you have dedicated to our work.
>
> Best regards,
>
> Authors of submission 9645

---

> > ### Comment · Reviewer_6uFu · 2025-11-27
> > **Final Reviewer Comment**
> >
> > I highly appreciate the detailed response to every question and comment posed.
> >
> > The improved draft does look better, and thank you for incorporating many of the suggestions and also for referencing other Reviewer suggestions that were included, too. Much appreciated.
> >
> > Some minor comments below are for the next iteration of the draft improvement (wrt current second version):
> >
> >
> >
> > B\1. L445-446: `LR drops to 4x`; should be `1/4` or `increases to`?
> >
> > B\2. Would recommend putting this remark into the paper, and thanks for the neat response: "Our analysis focuses on how reduced noise (e.g., via larger batch size) helps the optimizer move into deeper minima, conceptually different from the flat-minima perspective in fully interpolating regimes."
> >
> > B\3. Figure 6,8: could be made more legible by better styling choices:
> >   * Instead of varying transparency/opacity, could consider different line styles (`--`, `..`, `-x-`, etc.) for the same $T$ values while colors could represent the type of schedule (as now).
> >   * When more than 4-5 learning curves are being shown in the same plot, perhaps it is adequate to show only the smoothed loss and omit the true noisy curve for visual clarity.
> >   * For completeness, can still link the full plot in the Appendix.
> >
> > Thank you for the discussion and clarifications. Helped me understand the paper much better.
> >
> > The scores have been increased. All the best!

---

> > > ### Author Response · Authors · 2025-11-28
> > > **Thank you for positive feedback!**
> > >
> > > Dear Reviewer 6uFu,
> > >
> > > Thank you for your positive feedback and helpful suggestions! We truely appreciate you recognizing our revisions and increasing the scores.
> > > Regarding your minor comments:
> > > - B1: We have corrected L445-447 to state "LR drops to 1/4" or "LR drops to 1/16".
> > > - B2: We have incorporated the suggested remark in Footmark 8.
> > > - B3: We are stilling working on improving Figures 6 and 8 by using distinct line styles instead of transparency, showing only smoothed curves in main figures. These figures will be updated soon.
> > >
> > > The revisions are marked in **blue**. Thank you again for your thorough review and constructive feedback.
> > >
> > > All the best,
> > >
> > > Authors of submission 9645

---

### Official Review · Reviewer_fTYS · 2025-11-01

**Soundness:** 2
**Presentation:** 3
**Contribution:** 2
**Rating:** 4
**Confidence:** 4

**Summary:**

This paper aims to systematically study the geometrical evolution of the local loss landscape during LLM pre-training and correlate it with hyperparameter tuning strategies. The authors divide the process into two phases. In the first phase (early), authors reports a "from sharp to flat" evolutionary trend and, based on this, provides a geometrical explanation for the necessity of LR warmup via linear stability analysis. In the second phase (late), this paper argues that the landscape geometry is dominated by gradient noise. Through the continuous-time limit of Stochastic Differential Equations (SDEs) and the principle of free energy minimization, it reveals how batch size regulates a "depth-flatness" trade-off, thereby proposing a dynamic batch size scheduling strategy.

**Strengths:**

1. This paper intuitively documents and reports the "from sharp to flat" macro-dynamic trend in LLM pre-training, which is an important empirical finding that contradicts studies on small-scale models.

2. The proposed hyperparameter tuning strategies are logical, easy to implement, and experimentally shown to significantly improve training efficiency, contributing directly to reducing the cost of large model training.

**Weaknesses:**

1.  In Section 4, this paper attempts to explain the loss spikes and plateaus of the early training phase. The core argument is: first, it empirically observes that the initial landscape is very sharp (high Hessian eigenvalues), and then points out that a large LR on such a sharp landscape causes instability. To theoretically support this, the author borrows linear stability analysis from standard gradient descent (Lemmas 4.1 and 4.2). However, this analysis is strictly established on a **deterministic (noise-free), quadratic model centered around a local minimizer $\theta^{*}$** (Equation 4). **The key issue here is the applicability of this "extrapolation"**: While this paper does not claim the quadratic model *fits* the initial state, it *assumes* that the stability condition derived from this highly simplified, local-convergence model (i.e., $\eta < 2/\lambda_{max}(S)$) can effectively explain the dynamics of the earliest training phase (far from any minimizer, highly non-convex, and noisy). This is a strong assumption. The true early-stage dynamics are highly complex. Attributing the instability primarily to the simple linear interaction between LR and the local Hessian's max eigenvalue is likely an oversimplification, ignoring other non-linear or stochastic noise effects. Therefore, while the conclusion "high sharpness + high LR = instability" is intuitive and matches the data, using Lemmas 4.1 and 4.2 as its primary theoretical basis acts more as an insightful **analogy** than a rigorous proof for this specific phase. The validity of this explanation depends on the extent to which this local linear approximation dominates the early global dynamics, which is not sufficiently justified in this paper.

2.  The core theory in Section 5 (Prop 5.1 and Thm 5.1) relies on the SDE continuous-time limit, which assumes $\eta \to 0$. This contradicts modern LLM training practices (including the use of relatively large peak LRs in this paper's Section 4 experiments). Although the theory's key prediction (noise scale $\tau \propto \eta/B$) appears to match the experiments (Figure 8), this treats a heuristic approximation as a rigorous explanation.

3.  The study is limited to 93M and 170M parameter models under the LLaMA-2 architecture, which differs significantly from current mainstream model sizes. Whether its conclusions hold for much larger models remains unknown.

4.  This paper attributes the necessity of warmup to an external factor: the "landscape sharpness". However, a more direct and well-known explanation lies in the internal flaws of the Adam optimizer itself: its second-moment estimate ($v_t$) has high variance in the early stages, and its initial update degenerates into unstable "sign descent". The loss spikes observed in this paper are highly consistent with these known optimizer startup problems. This paper fails to clearly disentangle whether the observed instability originates from the landscape geometry or simply from the well-known startup deficiencies of the Adam optimizer.

**Questions:**

1.  Regarding the early-stage stability analysis: How do the authors justify that a noise-free, quadratic model (Eq 4), based on the neighborhood of a local minimizer, can effectively explain the instability phenomena in the earliest, far-from-equilibrium, and highly stochastic phase of training? Is there a more suitable theoretical model to describe this "chaotic initial" phase?

2.  Regarding the SDE limit and steady-state assumption: Given that LLM pre-training uses finite, large learning rates and is terminated long before reaching a theoretical steady state (stationary distribution), can the authors provide additional evidence or arguments to support the approximate validity of the SDE limit and free energy minimization theory in this scenario?

3.  Regarding the deeper reasons for blockwise heterogeneity: The authors attribute the ordering difference with Wang et al. to measurement methods and gradient sparsity. This raises a question: should we focus on the "intrinsic" Hessian geometry defined by the architecture, or the "effective" dynamic geometry jointly determined by data flow and the optimizer?

---

> ### Author Response · Authors · 2025-11-23
> **Response to Reviewer fTYS (1/2)**
>
> Thank you for your great efforts on the review of this paper. We will try our best to address your questions.
>
> **If you still have further concerns, or if you are not satisfied by the current responses, please let us know, so that we can update the response ASAP.**
>
> **Q1: On the validity of local quadratic model.** "this analysis is strictly established on a deterministic (noise-free), quadratic model ... The key issue here is the applicability of this 'extrapolation'" and "Is there a more suitable theoretical model to describe this 'chaotic initial' phase?"
>
> **A1**: Thank you for highlighting the limitations of our previous theoretical setup. We fully agree that our earlier analysis, which relied on a deterministic quadratic approximation, was too restrictive for capturing the dynamics in the early chaotic phase. In response, we have substantially revised our theory.
>
> First, we removed the local quadratic model assumption in the early phase. Instead of linear stability analysis under a quadratic model (Wu et al., 2018), we now adopt a more applicable framework based on Lyapunov stability, which characterizes the sensitivity of a dynamical system to small perturbations. Specifically, we consider two nearby trajectories, $\boldsymbol{\theta}\_k$ and $\boldsymbol{\tilde{\theta}}\_k$, and study their deviation $\boldsymbol{e}\_k:=\boldsymbol{\tilde{\theta}}\_k - \boldsymbol{\theta}\_k$. When the noise term $\boldsymbol{\xi}$ is set to zero, the deviation evolves as:
> $$
> \boldsymbol{e}_{k+1} = \boldsymbol{e}_k - \eta \mathbf{M} (\nabla L(\boldsymbol{\theta}_k + \boldsymbol{e}_k) - \nabla L(\boldsymbol{\theta}_k)) \overset{\text{(Linearization)}}{=} (\mathbf{I} - \eta \mathbf{M} \mathbf{H}(\boldsymbol{\theta}_k))\boldsymbol{e}_k.
> $$
> This linearization is accurate when $\boldsymbol{e}\_k$ remains small, and importantly, it does not require assuming a quadratic loss landscape. If $\lim\_{k\to\infty} \boldsymbol{e}\_k = \boldsymbol{0}$, the linear system is Lyapunov stable with respect to small perturbations.
>
> Second, following this theoretical revision, we have updated Lemma 4.1 and Lemma 4.2 in the main paper. Crucially, the original conclusions still hold under this framework; the results are now more robust theoretically.
>
> We thank the reviewer again for this helpful feedback, which has led to a clearer theoretical foundation in our revision.
>
> **Q2: On the validity of SDE modeling.** "The core theory in Section 5 (Prop 5.1 and Thm 5.1) relies on the SDE continuous-time limit, ...This contradicts modern LLM training practices." and "can the authors provide additional evidence or arguments to support the approximate validity of the SDE limit and free energy minimization theory in this scenario?"
>
> **A2:** Thank you. We address these concerns from three perspectives:
>
> - **Well-estabilished methodology.** Continuous-time models in the form of stochastic differential equations are a well-established tool to study discrete-time optimization process [1]. Similar SDE modelings have been adopted in many influential works [2, 3], and their validity has been empirically tested in [4]. Our use of an SDE limit follows standard practice in the community.
> - **Empirically validated in our setup.** Our predictions from the SDE model closely match real LLM training behavior. In particular, in Sec. 6, we compare the LR decay and BS ramping, and also evaluate their combined effect. We show that any scheduler (even those combining LR decay and BS ramping) that preserves the ratio $\eta / B$ produces nearly identical loss trajectories, exactly as predicted by our SDE-based theory. When $\eta / B$ differs, the trajectories diverge. This provides direct empirical validation of the SDE approximation in our regime.
> - **Intended role of the theory.** We also clarify that the goal of our theory is not to provide a full continuous-to-discrete convergence analysis for modern LLM training, which remains an open problem. Instead, our aim is to (1) explain the evolution of local landscape geometry during pre-training and (2) show how this informs hyperparameter tuning. Within this scope, the SDE model effectively captures the key mechanism, aligns with empirical observations, and guides the design of data-efficient BS schedules.
>
> [1] Helmke & Moore. Optimization and Dynamical Systems. Springer London 1994.
>
> [2] Li et al. Stochastic modified equations and adaptive stochastic gradient algorithms. ICML 2017.
>
> [3] Jastrzębski et al. Three factors influencing minima in sgd. 2017.
>
> [4] Li et al. On the Validity of Modeling SGD with Stochastic Differential Equations (SDEs). NeurIPS 2021.

---

> ### Author Response · Authors · 2025-11-23
> **Response to Reviewer fTYS (2/2)**
>
> **Q3: Suggestions for experiments on larger models and other architectures.** "The study is limited to 93M and 170M parameter models under the LLaMA-2 architecture, which differs significantly from current mainstream model sizes. Whether its conclusions hold for much larger models remains unknown."
>
> **A3**: Thank you for your suggestion. We have conducted additional experiments on larger models with 270M and 530M parameters, as well as on a different architecture `GPT-2`. Specifically, we evaluate the sharpness evolution in the early training iterations and validate our proposed BS scheduling method. We also verify the warmup-tuning recipe on the `GPT-2` architecture. These new results (see **Appendix D.2 and Fig. 9-10, 13**) demonstrate that our main conclusions remain consistent across larger scales and alternative architectures.
>
> **Q4: On the originality of loss spikes.** "its initial update degenerates into unstable 'sign descent'" and "This paper fails to clearly disentangle whether the observed instability originates from the landscape geometry or simply from the well-known startup deficiencies of the Adam optimizer."
>
> **A4**: Really insightful question. Yes, Adam might degenerate into Signum, but the "unstable" optimizer cannot explain the instability observed early in training, especially with shortened warmup and enlarged LR.
>
> To disentangle optimizer-induced instability from landscape-induced instability, we repeated the experiments using Muon, a substantially more stable optimizer than AdamW. In **Appendix D.3 and Fig. 14**, the loss spikes and plateaus consistently occurs under Muon when warmup is shortened or the peak LR is increased. This rules out the possibility that the behavior stems from AdamW’s startup issues.
>
> A more direct evidence comes from varying only the warmup length while fixing the peak LR at $2^{-7}$. In **Appendix D.3 and Fig. 15 (left)**, shorter warmup lengths lead to higher possibilities of loss spikes, and with no warmup the spike even occurs around initialization. This behavior is consistent with the sharp-to-flat dynamics. Early in training, the model resides in sharper regions of the landscape, where only sufficiently small LRs ensure stable updates. Therefore, warmup is needed to gradually increase the LR until the trajectory enters flatter regions that can tolerate larger LRs.
>
> Together, these results show that the observed instability originates from the training dynamics and landscape geometry, not merely from the optimizer.
>
>
> **Q5:** "The authors attribute the ordering difference with Wang et al. to measurement methods and gradient sparsity. This raises a question: should we focus on the 'intrinsic' Hessian geometry defined by the architecture, or the "effective" dynamic geometry jointly determined by data flow and the optimizer?"
>
> **A5**: A great question. Our answer is "it depends". When the goal is to accelerate training, we should consider the *effective dynamic geometry* shaped jointly by the data flow and optimizer. This geometry is estimated via gradients, directly influences optimization, and thus informs LR and BS choices. For example, Wang et al. (2025) used diagonal Fisher information to show that the embedding block is the flattest component and safely increased its LR by $10\times$, achieving faster pre-training. In contrast, the *intrinsic static geometry* determined by the architecture is more informative for understanding inductive biases or designing architectures, rather than guiding optimizations.

---

> ### Author Response · Authors · 2025-11-26
> **Looking forward your feedback**
>
> Dear Reviewer fTYS,
>
> We hope that our responses adequately address your concerns. As the deadline of this discussion phase is approaching, we warmly welcome further discussion regarding any additional concerns that you may have, and we sincerely hope you can reconsider the rating accordingly.
>
> Thank you for the time and appreciation that you have dedicated to our work.
>
> Best regards,
>
> Authors of submission 9645

---

### Author Response · Authors · 2025-11-23
**Global Response**

We thank all reviewers for their constructive feedback. We have substantially revised the paper to strengthen the theory, broaden the empirical validation, and clarify the presentation. Below is a short summary of the main concerns and our changes.

| Aspect | Main concern | Our response / key changes |
| -- | -- | -- |
| **Theory** | Validity of the local quadratic assumption. | We **removed** the local quadratic assumption in Sec. 4 and replaced it with a **Lyapunov-based stability analysis** on linearized dynamics between nearby trajectories. This preserves the original conclusions while applying to general nonlinear loss landscapes. |
| **Theory** | Validity of the SDE modeling. | We clarify that the SDE modeling is a **standard tool** to analyze training dynamics; we use it to explain empirical behavior and motivate BS schedules. Developing stronger theory is important, but is a **seperate interest.** |
| **Experiments**  | Limited experimental scope.   | We added **new experiments** (**Appendix D.2–D.3**) on **larger models** (270M, 530M), **GPT-2**, and **alternative optimizers** (Muon, Lion, Adam-mini), which further confirm our main conclusions. |
| **Experiments**  | Interaction between LR decay and BS ramping.   | We systematically study the **interaction between LR decay and BS ramping** (**Fig. 8**): any schedule (including hybrid ones) that preserves $\eta/B$ produces nearly identical late-phase loss curves, confirming that the noise scale $\tau \propto \eta/B$ governs the dynamics.  |
| **Experiments**  | Effects of timing and number of BS ramping.   | We add **new experiments** on the **timing and number of BS ramps** (**Fig. 6–7**): all BS ramping schedules collapse onto the same loss trajectory, and under a fixed token budget, **late BS ramping** performs best, with diminishing returns from multiple ramps. |
| **Novelty**      | Limited novelty over prior CBS-inspired BS scheduling. | We show that all BS ramping schedules converge to the same loss trajectory, and that delaying the BS ramp improves final loss under a fixed token budget. This leads to our design principle II: **ramp BS only very late**, once loss reduction becomes marginal, which CBS theory and BS-warmup heuristics do **not** predict. |
| **Presentation** | Terminology, structure, and limitations. | (1) We added a **terminology table** (**Appendix A**) for sharpness, flat/sharp minima, and wide/deep basins. (2) Sec. 5 is reorganized into a clearer narrative: empirical observations → SDE-based interpretation → **two design principles** for BS scheduling. (3) We expanded the **Limitations** section (**Appendix B**) to discuss untested hyperparameters and architectural coverage. |

We hope these revisions clarify our contributions, especially the **late BS ramping principle in the data-limited regime** and the **sharp-to-flat early-phase dynamics with proportional warmup scaling**, and address the main concerns raised by the reviewers.

---

### Author Response · Authors · 2025-12-02
**A Short Summary of Rebuttal**

Dear AC,

We acknowledge the recent reviewer-information leakage and the resulting changes to the review process, and we appreciate the additional effort required from you.

 We believe our rebuttal has substantially addressed their main concerns:
- Reviewer eiwb & 6uFu keeps **positive** ratings, with Reviewer 6uFu highly appreciating our revision and increasing their score from **4 to 6**.
- For Reviewer fTYS & G3Yh, we provided point-by-point responses to all comments, although they did not respond yet.

To assist you in forming recommendation, we provide a brief summary of our rebuttal below.

**1. Validty of Theoretical Simplifications.**

Reviewers highlighted the need to strengthen theoretical foundations, specifically questioning the validity of the local quadratic assumption, SDE modeling applicability, etc.
- **Removed the local quadratic assumption**. We replaced the restrictive quadratic approximation in Sec. 4 with a *Lyapunov-based stability analysis* that characterizes dynamics between nearby trajectories. This framework does not require quadratic loss structure, applies to general nonlinear dynamics, and preserves our original conclusions (sharp-to-flat early-phase dynamics).
- **Clarified SDE modeling’s role and validity**.
    - We emphasize that SDEs are a standard tool in optimization dynamics and clarify their purpose: to explain landscape evolution and motivate practical scheduling, *not* to provide full discrete-to-continuous convergence proofs.
    - We validate SDE predictions empirically: New experiments (revised Fig. 8) show that any schedule (LR decay, BS ramping, or hybrid) preserving the noise scale *η*/*B* produces nearly identical loss curves, directly confirming the SDE model’s relevance to real LLM training regime.
- **Justified theoretical simplifications**: We clarified that the noise covariance assumption Σ∝1/*B* derives from discrete-time variance analysis (Lemma C.4).

**2. Scope of Experimental Results**

Reviewers noted limitations in the original experimental scope (93M/170M LLaMA-2 models, AdamW only) and requested validation across larger scales, alternative architectures, and optimizers.
- **Larger model sizes**: Added experiments on 270M and 530M parameter models (Fig. 13). Results confirm that our BS scheduling strategy becomes *more* effective with scale.
- **New architectures/optimizers**: Validated key findings on GPT-2 (architecture) and Lion, Muon, Adam-mini (optimizers) (Fig. 9–12, 14). Critical results (sharp-to-flat early dynamics, BS scheduling gains) hold across all tested setups.
- **Disentangled optimizer vs. landscape effects**: Using Muon (a more stable optimizer than AdamW), we show loss spikes persist with shortened warmup/large LR, ruling out AdamW’s startup issues as the root cause (Fig. 14–15).

**3. Hyperparameter Interaction and Scheduling Principles**

Reviewers requested deeper analysis of LR decay/BS ramping interactions and more concrete guidance on BS ramping timing/number.
- **Systematically studied LR decay × BS ramping**: New experiments (Fig. 8) demonstrate that dynamics are governed by *η*/*B*: Hybrid schedules (e.g., LR decay + BS ramping) preserving this ratio perform identically to LR decay and BS ramping alone.
- **Clarified BS ramping principles**:
    - *Timing*. All BS ramping schedules converge to the same loss trajectory (Fig. 6 left), but *late ramping* yields optimal performance under fixed token budgets (Fig. 6 right), a key insight not predicted by CBS theory.
    - *Number of ramps*. Multiple ramps offer diminishing returns (Fig. 7).
- **Concretized LR warmup guidance**: Revised Fig. 4 shows optimal warmup length scales *proportionally* with peak LR within $2^{-11}$ to $2^{-8}$ LR range and validates this on GPT-2 (Fig. 10), providing actionable tuning for practitioners.

**4. Novelty and Distinction from Prior Work**

Reviewer G3Yh questioned novelty relative to prior SDE/CBS work (Jastrzębski et al. 2018, Merrill et al. 2025) and LR warmup literature (Kalra & Barkeshli 2024).
- **Distinction from SDE/CBS theory**:
    - *Different goal*. Jastrzębski et al. (2018) focus on generalization/width of minima controlled by *η*/*B*; we use SDEs to derive a *data-efficient BS scheduler* for LLM pre-training.
    - *Different method*. *CBS-based methods (Merrill et al. 2025)* ramp BS according to CBS evolution to optimize training time while preserve data efficiency; our *late ramping* principle optimizes loss under fixed tokens, suggesting ramping the BS until the very late phase of training, even when CBS theory permits early ramping (a contradictory insight).
- **Novelty in LR warmup**: Kalra & Barkeshli (2024) study small-scale models and do not show proportional warmup-LR scaling or sharp-to-flat dynamics in LLMs. We provide the first direct visualization of sharpness evolution in realistic LLM pre-training and link it to a actionable warmup rule.

Thank you for your time.

Best regards,

Authors of submission 9645

---

### Meta-Review · Area_Chair_s23f · 2026-01-12

**Summary:**

The problem studied is interesting and meaningful, but the main concern is that the novelty is limited. The empirical findings in this paper are quite well-known. The authors then mention that "theory is not the main contribution" in the response to Reviewer G3YH. Unfortunately, this leaves not much new beyond what is already known. Therefore, although this paper may serve as a reminder to practitioners with some useful guidance, as a research paper the level of new contributions is insufficient for a top-tier machine learning conference.

**Reviewer Concerns:**

The concerns about novelty are still outstanding.

**Reviewer Scores:**

I think Reviewer G3Yh would not upgrade the score and maintain a negative opinion.

---

### Decision · Program_Chairs · 2026-01-26

Reject